# Chemo- and optogenetic activation of hypothalamic *Foxb1*-expressing neurons and their terminal endings in the rostral-dorsolateral PAG leads to tachypnea, bradycardia, and immobility

Reto B Cola[1]*, Diana M Roccaro-Waldmeyer[1], Samara Naim[1], Alexandre Babalian[1], Petra Seebeck[2], Gonzalo Alvarez-Bolado[3], Marco R Celio[1]*

[1]Anatomy and program in Neuroscience, Faculty of Science and Medicine, University of Fribourg, Fribourg, Switzerland; [2]Zurich integrative Rodent Physiology (ZIRP), University of Zürich, Zürich, Switzerland; [3]Institute of Anatomy and Cell Biology, University of Heidelberg, Heidelberg, Germany

*For correspondence:
reto.cola@uzh.ch (RBC);
marco.celio@unifr.ch (MRC)

**Competing interest:** The authors declare that no competing interests exist.

**Abstract** *Foxb1*-expressing neurons occur in the dorsal premammillary nucleus (PMd) and further rostrally in the parvafox nucleus, a longitudinal cluster of neurons in the lateral hypothalamus of rodents. The descending projection of these *Foxb1*+ neurons end in the dorsolateral part of the periaqueductal gray (dlPAG). The functional role of the *Foxb1*+ neuronal subpopulation in the PMd and the parvafox nucleus remains elusive. In this study, the activity of the *Foxb1*+ neurons and of their terminal endings in the dlPAG in mice was selectively altered by employing chemo- and optogenetic tools. Our results show that in whole-body barometric plethysmography, hM3Dq-mediated, global *Foxb1*+ neuron excitation activates respiration. Time-resolved optogenetic gain-of-function manipulation of the terminal endings of *Foxb1*+ neurons in the rostral third of the dlPAG leads to abrupt immobility and bradycardia. Chemogenetic activation of *Foxb1*+ cell bodies and ChR2-mediated excitation of their axonal endings in the dlPAG led to a phenotypical presentation congruent with a 'freezing-like' situation during innate defensive behavior.

## eLife assessment

This paper describes **valuable** results from studies investigating circuits in the brain that underlie behavioral responses in fearful situations. The authors identified a role for a class of neurons that are sufficient to cause these stereotyped behaviors including freezing behaviors. These **solid** studies increase our understanding of brain pathways regulating these types of behaviors.

## Introduction

Mapping the circuits that form the basis of specific behaviors is essential to understand the brain and is an essential task for the neuroscience of our time. The circuitries that underlie fear are important in the clinic and consequently have become a hotspot of neuroscientific research in the last few years (*Tovote et al., 2015*; *Mobbs et al., 2020*; *Silva and McNaughton, 2019*). Here, we have used chemogenetics and optogenetics to show that the activation of a specific, well delimited neuronal group in the hypothalamus leads to innate defensive behavior, including tachypnea, immobility, and bradycardia.

During hypothalamic development, two neuronal migratory streams expressing the transcription factor *Foxb1* form the nuclei of the mamillary region as well as one nucleus of the lateral hypothalamus (LHA). One of these streams forms most of the mamillary body while the other forms the dorsal premammillary nucleus (PMd) and the *Foxb1*-expressing neurons of the parvafox nucleus (parvafox-$^{Foxb1}$) in the LHA (*Alvarez-Bolado et al., 2000*). The ventrolateral part of the PMd (vlPMd) projects to the dorsolateral periaqueductal gray (dlPAG; *Canteras and Swanson, 1992*) while the parvafox-$^{Foxb1}$ axon terminals occupy a region straddling the dlPAG and the lPAG (*Bilella et al., 2016*). Solid evidence indicates that the dlPAG is associated with innate defensive responses like 'freezing' and escape behavior (*Canteras and Goto, 1999*; *Vianna et al., 2001*; *Bittencourt et al., 2004*; *Deng et al., 2016*; *Kunwar et al., 2015*; *Souza and Carobrez, 2016*; *Evans et al., 2018*).

In addition to inputs from *Foxb1*+ cells of the hypothalamus, the dlPAG also receives afferences from neurons of the ventromedial hypothalamus (VMH) expressing steroidogenic factor 1 (*SF1*+). The optogenetic activation of their cell bodies in the VMH (*Kunwar et al., 2015*) and of their terminal endings in the dlPAG (*Wang et al., 2015a*) leads to immobility. As the stimulation intensity increases, immobility is followed by bursts of activity (running and jumping; *Kunwar et al., 2015*). This behavioral change is ascribed to the co-activation of another axon collateral of the *SF1*+ neurons, innervating the anterior hypothalamic nucleus (AHN; *Wang et al., 2015a*). As animals remain immobile, increasing the intensity of the stimulation of *SF1*+ cell bodies in the VMH leads to a change of the cardiovascular responses from tachycardia to bradycardia (*Wang et al., 2015b*).

The third afference to the dlPAG (*Schenberg et al., 2005*) stems from medially located glutamatergic neurons of the superior colliculus (mSC), which is responsive to innately aversive looming stimuli (*Evans et al., 2018*). mSC neurons integrate threat evidence and pass the information through a synaptic threshold to the glutamatergic dlPAG neurons that initiate escape (*Evans et al., 2018*).

We set out to establish the function of two distinct neuronal groups identifiable through *Foxb1* expression, the parvafox$^{Foxb1}$ of the LHA and the PMd$^{Foxb1}$ of the hypothalamic mamillary region, both projecting to specific regions of the PAG (*Canteras and Swanson, 1992*; *Bilella et al., 2016*). To that end, we used the specific expression of Cre recombinase in Foxb1$^{tm1(cre)Gabo}$ mice as a tool.

In our study (*Figure 1*), projection-unspecific chemogenetic activation of the *Foxb1*+ cell bodies of the PMd and of the parvafox$^{Foxb1}$ led to tachypnea, while selective optogenetic activation of theire terminal endings in the rostral part of the dlPAG provoked immobility, accompanied by bradycardia.

## Results

### Chemogenetic modulation of the *Foxb1* neurons increases breaths per minute (BPM) without altering tidal volume (TV)

The activation of the hM3Dq (activating DREADD receptors) expressed in the parvafox$^{Foxb1}$ by intraperitoneal clozapine or CNO injection led to changes in several parameters of respiration consistent with an increased respiratory effort *Figure 2*; breaths per minute (BPM), $F_{(2,26)}$=6.061, p=0.00691, g=; inspiratory time (IT), $F_{(2,26)}$ = 5.831, p=0.00809; total time (TT) $F_{(2,26)}$=4.655, p=0.01973; minute volume adjusted for bodyweight (MVadjPerGram), $F_{(2,26)}$=3.998, p=0.03265; peak inspiratory flow (PIFadj), $F_{(2,26)}$ = 4.284, p=0.03439; all reported results are obtained by a 3x3 mixed-design ANOVA for the 'Condition' factor, p values are Huynh-Feldt sphericity corrected. Namely, we observed a significant increase in three parameters within saline, clozapine and CNO injected hM3Dq animals: (1) BPM (*Figure 2a–I*; saline vs. clozapine: $t_{(5)}$ = –3.199, p=0.024, g = –1.380 (large); saline vs. CNO: $t_{(5)}$ = –2.916, p=0.033, g = –1.192(large); clozapine vs. CNO: $t_{(5)}$ = 3.069, p=0.028, g=0.226(small)), (2) MVadjPerGram: (*Figure 2a–VI*; saline vs. clozapine: $t_{(5)}$ = –2.790, p=0.038, g=–0.974(large); saline vs. CNO: $t_{(5)}$ = –2.600, p=0.048, g = –0.928(large)) and (3) PIFadj (*Figure 2a–VII*; saline vs. clozapine: $t_{(5)}$ = –2.726, p=0.041, g = –1.217(large); all reported results are post-hoc two-tailed paired student's t-Tests; g values represent the effect size according Hedge's g corrected for paired data, the magnitude of the effect size is provided in parentheses [i.e. negligible, small, medium, or large]). Remarkably, the tidal volume normalized for bodyweight (TVadjPerGram; *Figure 2a-IV*) was not altered by the intervention (3x3 mixed-design ANOVA: $F_{(2,26)}$ = 1.165, p=0.322). Furthermore, since minute volume equals the tidal volume multiplied by BPM, it follows that, the principal factor explaining the changes in MVadjPerGram was the increased BPM. The inspiratory time (IT, *Figure 2a–II*; ; saline vs. clozapine: $t_{(5)}$ = 3.296, p=0.022, g=1.170[large]) and the total respiratory time (TT, *Figure 2a-IV*; saline vs. clozapine::

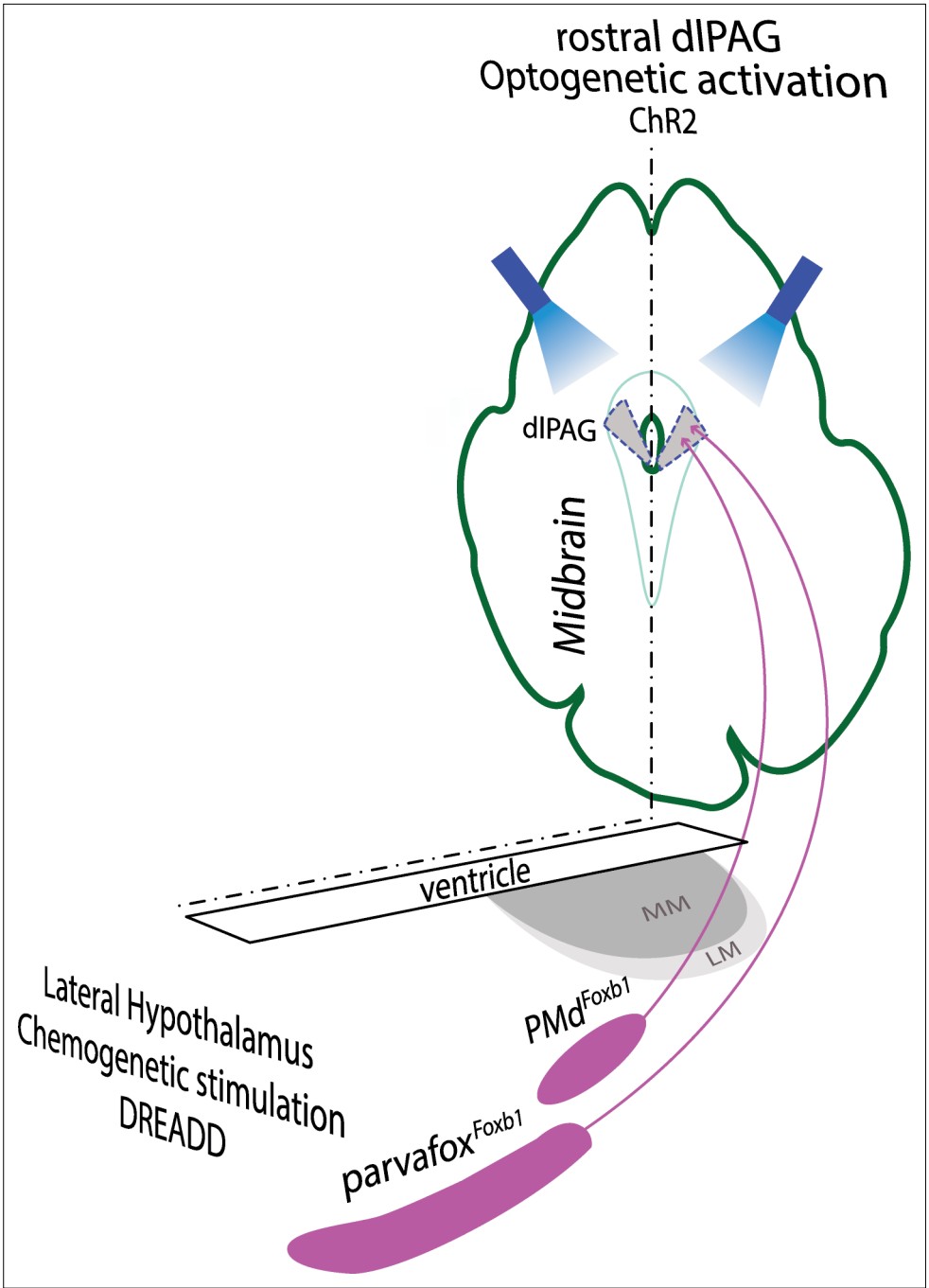

**Figure 1.** Left part: Schematic drawing of a horizontal section of the hypothalamus in which the Foxb1-neurons of the parvafox and of the PMd are located (modified after *Alvarez-Bolado et al., 2000*). Their chemogenetic (DREADD) stimulation leads to an increase in breaths / minutes. Right part: dorsolateral PAG (dlPAG), seen in a cross-section of the midbrain. Axon terminals of parvafox[Foxb1] and PMd[Foxb1] converge in the rostral part of the dlPAG. Optogenetic activation of these terminals lead to immobility and bradycardia.

$t_{(5)}$ = 2.683, p=0.043, g=1.140[large]) decreased, while however the expiratory time (*Figure 2a–III*) remained unaffected (ET; mixed-design ANOVA: $F_{(2,26)}$ = 1.656, p=0.217). Therefore, the increase in BPM was achieved by selectively shortening the IT through an increase in PIFadj, while expiratory parameters (e.g. PEFadj (*Figure 2a–VIII*)) did not achieve statistical significance level and therefore do not seem to have contributed significantly to the increase in BPM.

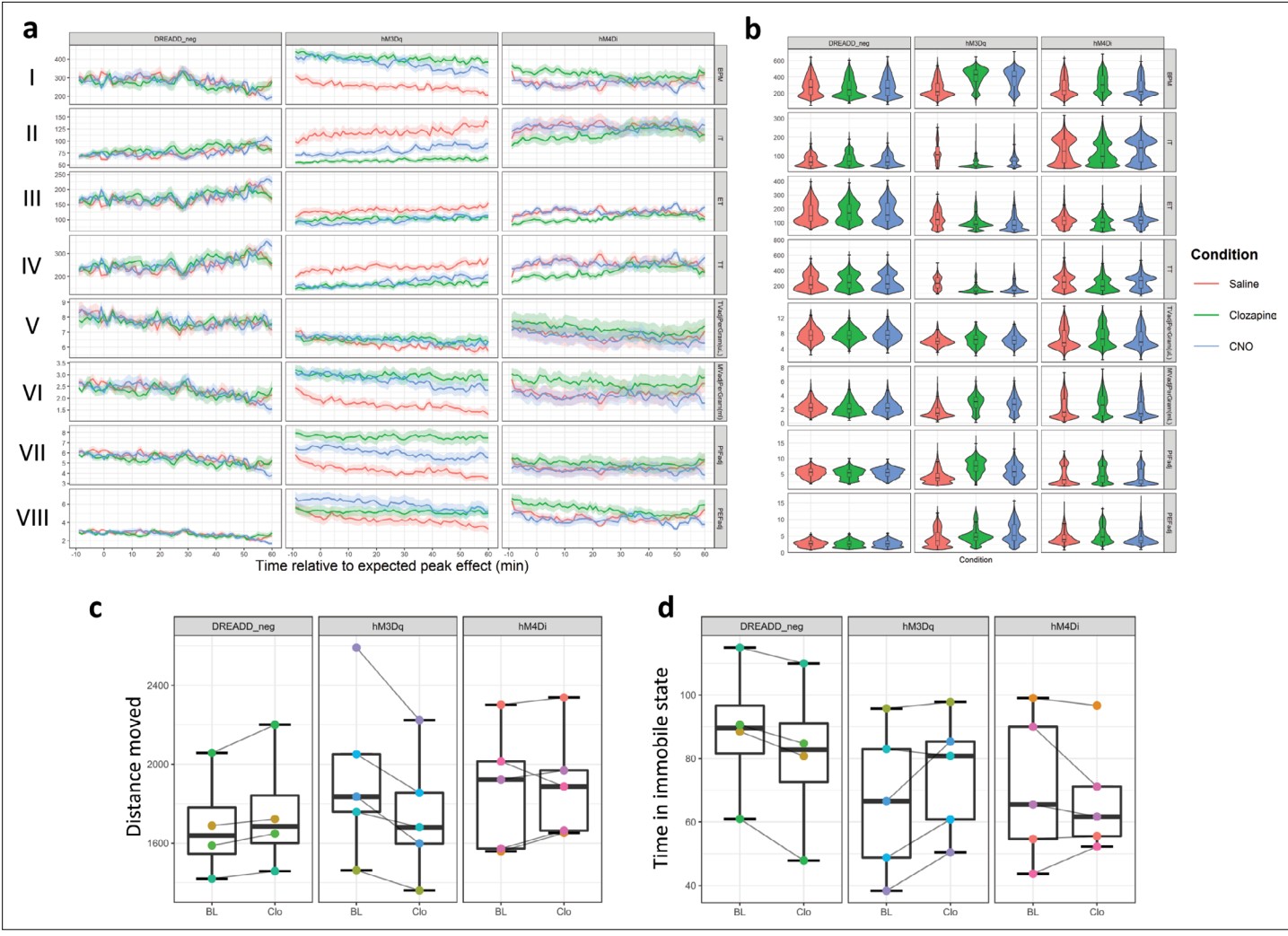

**Figure 2.** Chemogenetic activation of lateral hypothalamic Foxb1-expressing neurons alters multiple respiratory parameters. (**a**) Data across all DREADD WBP recordings was plotted as averaged line plots for each animal group (DREADD_neg, hM3Dq and hM4Di) and condition (Saline (red), Clozapine (green) or CNO (blue) i.p.injection). Each animal was recorded three times within each condition (nine recordings in total). Respiratory parameters do not differ between condition in DREADD_neg animals. Injection of CNO or Clozapine in hM3Dq mice significantly alters several respiratory parameters. The effects of Clozapine and CNO injections on respiratory parameters largely overlap and clearly separate from the effects saline injections. In hM4Di animals, more variability within respiratory parameters compared to DREADD_neg controls, however, no statistically significant effect is detected. Data shows the condition mean and standard error of the mean (s.e.m.) (**b**) Violin plots with integrated boxplots for each group and condition. The width of the violin plot represents the data distribution density. The boxplot 's lower and upper limits represent the 25% quantile and 75% quantile, respectively. The horizontal bar inside the boxplot represents the median. The whiskers of the boxplot display 1.5 x the interquartile range. (**c**) Comparison of gross locomotion as assessed by an open field test. hM3Dq animals show statistically significant reduction in distance moved, while there are no differences observed in the DREADD_neg and hM4Di animals. It is important to note, that the effect size for this reduction in distance moved is small. (**d**) A statistically significant reduction in time spent in immobility is observed in DREADD_neg animals, however, the effect size is negligible. Time spent in an immobile state does not differ in hM3Dq and hM4Di animals. Track visualization with underlying density maps and zone visit diagrams during open field tests for all DREADD mice can be found in ***Supplementary files 1 and 2***. I = BPM (Breaths per minute); II = IT (Inspiratory time); III = ET (Expiratory time); IV = TT (Total respiratory cycle time); V = TVadjPerGram (Tidal volume normalized to bodyweight in microliters per gram); VI = MVadjPerGram (Minute volume normalized to bodyweight in milliliters per gram); VII = PIFadj (Peak inspiratory flow); VIII = PEFadj (Peak expiratory flow); CNO: clozapine-N-Oxide. Number of mice per condition: DREADD_neg n=4, hM3Dq n=5, hM4Di n=5.

The online version of this article includes the following figure supplement(s) for figure 2:

**Figure supplement 1.** Representative examples of DREADD expression and cFos staining in four mice of each condition (DREADD_neg, hM3Dq, and hM4Di) used for chemogenetic experiments.

It is important to note that none of the parameters were altered to a level of statistical significance neither in the control group (DREADD_neg) nor in the inhibitory DREADD group (hM4Di). This indicates that the effect was caused by neuronal excitation of the parvafox$^{Foxb1}$ and not by an inherent effect of the injected substances (i.e. clozapine or CNO). This conclusion is further strengthened by the increase of c-Fos immunoreactivity evident in the parvafox nucleus of hM3Dq expressing mice, but not in those of hM4Di expressing mice nor in control mice (*Figure 2—figure supplement 1*).

To test for a potential bias of respiratory recordings by alterations in gross locomotor activity, an open-field test was performed (*Figure 2c and d*). Global hM3Dq activation of the parvafox$^{Foxb1}$ decreased the distance moved to a level of statistical significance, however, with only a small effect size ($t_{(4)}$ = 3.774, p=0.02, g=0.225(small)). The time spent in an immobile state was not affected by chemogenetic activation of the parvafox$^{Foxb1}$ ($t_{(4)}$ = 2.257, p=0.087). Activation of the inhibitory hM4Di receptors did not affect gross locomotor activity distance moved: $t_{(4)}$ = –0.697, p=0.524; time spent in immobile state: $t_{(4)}$ = 0.695, p=0.525. Injections of clozapine in control mice (DREADD_neg) did not alter distance moved ($t_{(3)}$ = –2.708, p=0.073), but lead to a statistically significant decrease of time spent in an immobile state, however, with a negligible effect size $t_{(3)}$ = 4.336, p=0.027, g=0.081 (negligible). Thus, despite reaching statistical significance for distance moved in hM3Dq animals and for time spent in an immobile state in DREADD_neg animals, the magnitude of the effect size of chemogenetic modulation of the parvafox$^{Foxb1}$ on gross locomotor activity is only small and negligible, respectively, and is unlikely to explain the large size of the effects observed in the alterations of respiratory patterns.

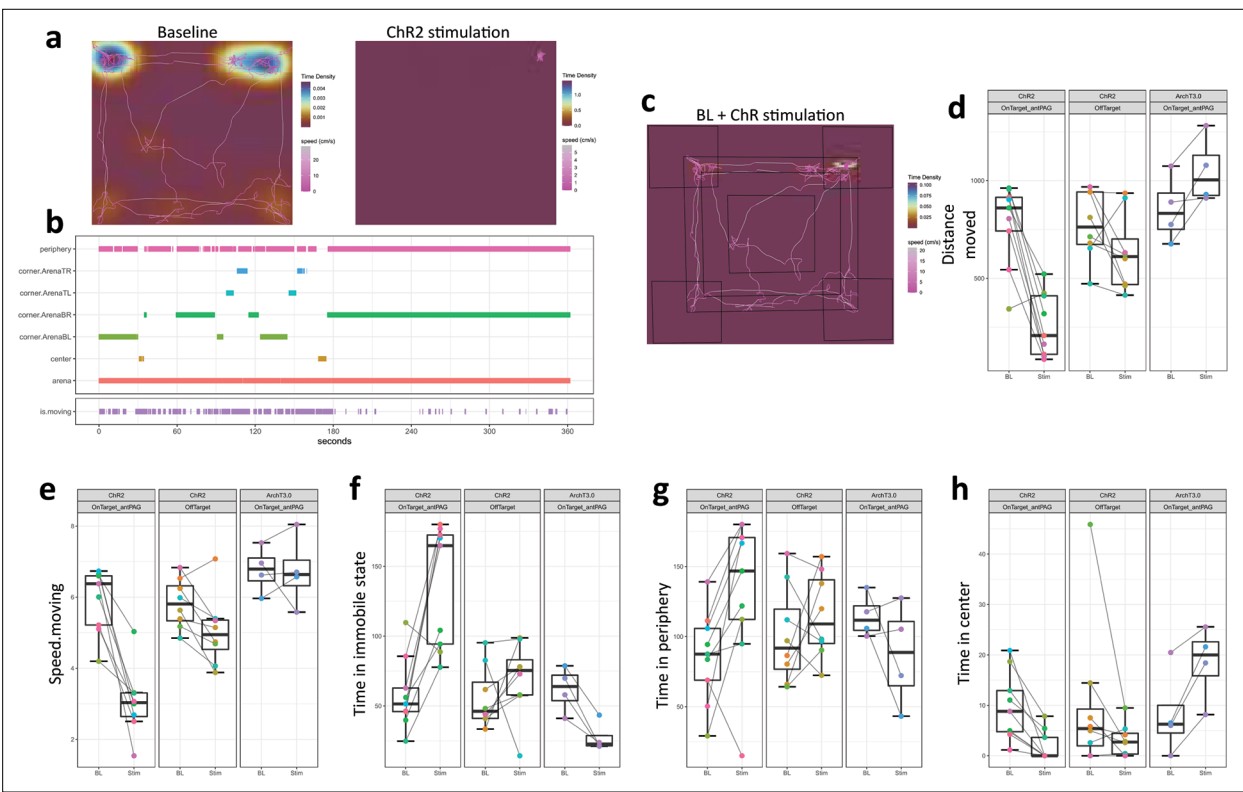

**Figure 3.** Optogenetic activation of hypothalamic Foxb1+ terminals in the rostral dlPAG induces freezing-like behavior. (**a**) A representative example of track visualizations with underlying density maps for a ChR2-expressing Foxb1-Cre mouse belonging to the 'OnTarget_antPAG' group during 3 min of baseline (left) and 3 min of optogenetic stimulation (right). Note that the mouse remained immobile for the entire duration of the stimulation period. The color coding scales are not fixed between the two conditions. Open field arena dimensions are 40x40cm. Track visualization with underlying density maps for all other mice can be found in *Supplementary file 3*. (**b**) A representative example of a zone visit diagram taken from the same recording as the track visualizations in (**a**). Note, that the zone transitions stop completely after the onset of optogenetic stimulation. Zone visit diagrams for all other mice can be found in *Supplementary file 4*. (**c**) A visual representation of the arena partitioning into the different zones (i.e. arena, periphery, center, and four corners). (**d-h**) Comparison of multiple parameters extracted from the open field experiments. Optogenetic activation of parvafox$^{Foxb1}$ terminals in the PAG reduces distance moved (**d**), speed.moving (**e**), and time in center (**h**), while time spent immobile (**f**) and time in periphery (**g**) increases. Optogenetic silencing of parvafox$^{Foxb1}$ terminals in the PAG induces opposing effects.

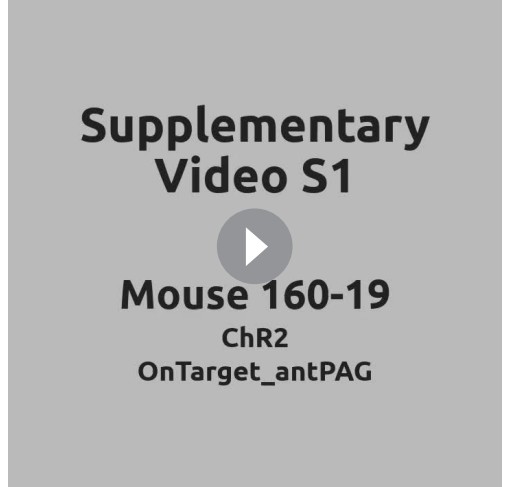

**Video 1.** A representative movie for a ChR2-stimulated mouse that was still capable of moving its head as a sign of attentive behavior towards its surroundings, while all four limbs remained largely immobile. The video shows the time window from 1 min before the onset of light stimulation, 5 min of light stimulation, and 1 min following light stimulation offset. The time window of light stimulation is marked by the 'Light on' indicator on the top right. The video is played at 5 × of the recording speed.

https://elifesciences.org/articles/86737/figures#video1

## Optogenetic modulation of *Foxb1* terminals in the dlPAG induces immobility

Respiratory results aside (see previous paragraph), the most prominent effect observed in the optogenetic experiments was the immobility ('freezing behavior') displayed by the group of ChR2-expressing mice during photoactivation of the hypothalamic *Foxb1*⁺ axonal projections to the rostral part of the dlPAG (*Figure 1*; Bregma –3.40/–4.04; *Figure 3*). These mice exhibited a short-latency immobility response immediately after onset of the photostimulation with blue light (7–15 mW, burst of 500ms duration with an intraburst frequency of 30 Hz (5ms pulse duration) and an interburst interval of 500ms). A decrease of locomotor activity was observed in 13 out of 16 ChR2-expressing mice (*Figure 3a–c*), of which eight mice showed almost complete absence of locomotion. Three of these mice were still capable of moving their heads as a sign of attentive behavior towards its surroundings (158/19, 160/19, and 35B-20), while all four limbs remained largely immobile (*Video 1*). Five other mice (162/19, 106/21–10, 34/21–7, 34/21–10, and 35E-20) displayed an even greater effect and remained completely immobile (including head activity) during almost the entire duration of the LEDon period (*Video 2*). No activity bursts were observed, neither during nor after the photostimulation period. Increasing the intensity of the stimulation from 70 to 222 mW (measured at patchcord) using a laser beam did not trigger motor activities.

To quantify the locomotive behavior induced by optogenetic modulation, we performed open field experiments, where mice were recorded for 3 min without photo-stimulation (BL) and 3 min with photo-stimulation (Stim).

Since we observed different degrees of immobility displayed by ChR2-expressing mice, we allocated each mouse to either an 'OnTarget_antPAG' or an 'OffTarget' group based on the histologically confirmed optic fiber placement. OnTarget_antPAG animals had the tip of the optic fiber implant located above the dlPAG at an anterior-posterior level AP-4.04mm (from bregma) or proximal. The OffTarget group contains animals with fiber tips located below (i.e. ventral to) the dlPAG and/or located more distal than AP –4.04 mm. Two-tailed paired Welch's *t*-Tests revealed significant effects of photoactivation in the following parameters in OnTarget_antPAG mice with ChR2 expression in the ventral-posterior hypothalamus (*Figure 3a–h*): Distance moved ($t_{(8)}$ = 5.934, p<0.001, *g*=2.506 [large]), speed.moving: ($t_{(8)}$ = 6.003, p<0.001, *g*=2.612 [large]), time in immobile state: ($t_{(8)}$ = –4.704, p=0.002, *g* = –1.976 [large]), time in center ($t_{(8)}$ = 4.108, p=0.003, *g*=1.178 [large]) and time in periphery ($t_{(8)}$ = –2.841, p=0.022, *g*=–0.903 [large]). All these parameters had a trend in the same direction in the OffTarget group; however, none besides speed.moving were altered to a level of statistical significance (*Figure 3d–h*): Distance moved ($t_{(7)}$ = 1.725, p=0.128), speed.moving: ($t_{(7)}$ = 2.895, p=*0.023*, *g*=0.779 [medium]), time in immobile state: $t_{(7)}$=–0.996, p=0.352, time in periphery ($t_{(7)}$ = –0.899, p=0.398). Besides speed.moving: ($t_{(3)}$ = 0.096, p=0.929), photoinhibition in ArchT3.0 OnTarget_antPAG mice consistently changed locomotor parameters in a direction opposite to the ChR2 mice (*Figure 3a–h*): Distance moved ($t_{(3)}$ = –3.176, p=0.050, *g* = –0.832 [large]), time in immobile state: ($t_{(3)}$ = 5.369, p=0.013, *g* = –1.712 [large]), time in center ($t_{(3)}$ = –4.590, p=0.019, *g* = –0.885 [large]) and time in periphery ($t_{(3)}$ = 1.367, p=0.265).

A comment about the specificity of the optogenetic results: the large volume of virus injected (200 nl) and the highly penetrant and strongly expressing viral serotype of the vector used (AAV5) led

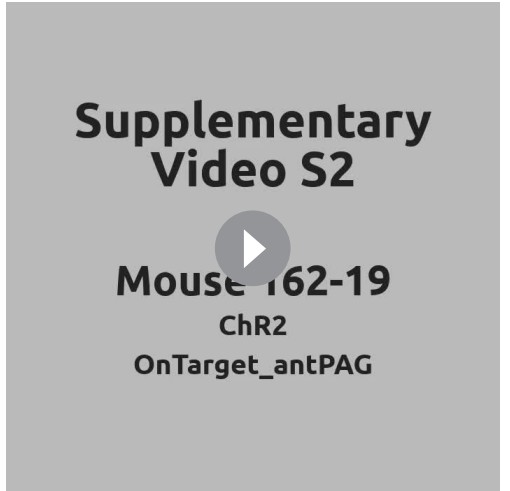

**Video 2.** A representative movie for a ChR2-stimulated mouse that was completely immobilized during optogenetic activation of the parvafox$^{Foxb1}$ terminals in the dlPAG. The video shows the time window from 1 min before the onset of light stimulation, 5 min of light stimulation, and 1 min following light stimulation offset. The time window of light stimulation is marked by the 'Light on' indicator on the top right. The video is played at 5 x of the recording speed.

https://elifesciences.org/articles/86737/figures#video2

to expression of ChR2 not only (as intended) in the parvafox$^{Foxb1}$ neurons but also in other *Foxb1*-expressing hypothalamic neurons in the neighboring mammillary region.

However, of all the mammillary nuclei only the dorsal premammillary nuclei (PMd) send axons to the rostral dlPAG (*Vertes, 1992*; *Meller and Dennis, 1986*; *Canteras and Swanson, 1992*), where the tip of the glass fibers for the optogenetic activation were implanted. In our experiments, the terminals from the PMd$^{Foxb1}$ were therefore co-activated with those of the parvafox$^{Foxb1}$. Projections to the rostral dlPAG from other mamillary neurons are very scarce (*Allen and Hopkins, 1990*; *Shen, 1983*; *Beart et al., 1990*; *Cruce, 1977*; *Canteras et al., 1992*).

In summary, optogenetic activation of *Foxb1*$^+$ terminals originating from the hypothalamic parvafox nucleus and/or PMd and projecting to the anterior PAG induces a state of immobility and hypoactivity. Such an activation namely decreases the distance moved, the speed during locomotor periods (speed.moving), the time spent in the center region of the arena, and increases the time spent in an immobile state and the time spent in the periphery of the arena. Stimulation at more posterior positions along the PAG and/or ventral to the dlPAG columns are inefficient in inducing this behavior. The observed effect is bidirectional, in the sense that inhibition of the same terminals in the anterior PAG induces increased distance moved, and time spent in the center of the arena, while decreasing the time spent in an immobile state.

## scRNA seq dataset reveals distinct *Foxb1* expression in the PMd

Our results to this point indicated that a population of *Foxb1*-expressing neurons in the PMd induces immobility in mice. In contrast to this finding, a cholecystokinin (*Cck*)-expressing population of neurons in the PMd has been shown to induce escape behavior in mice (*Wang et al., 2021*). We hypothesized that the *Cck*-expressing and the *Foxb1*-expressing PMd neurons are distinct, separate neuronal groups regulating opposite behaviors. To validate this hypothesis, we performed a reanalysis of previously published single-cell RNA sequencing datasets focusing on the murine posterior ventral hypothalamus (*Mickelsen et al., 2020*; *Figure 4*).

After quality control, normalization, and integration of all four datasets (from two female and two male mice), k-nearest neighbor clustering resulted in 24 distinct cell types detected in the ventral-posterior hypothalamus (*Figure 4a*). Plotting the expression levels of a set of PMd markers identified by Mickelsen et al. (i.e. *Cck, Foxb1, Synpr, Dlk1, Ebf3*, and *Stxbp6*) onto the UMAP plot identified cluster 9 as the PMd (*Figure 4b*). The cluster identity was further confirmed by calculating differentially-expressed genes between the PMd cluster and the other *Cck*- and *Foxb1*-expressing clusters (i.e. clusters 5, 7, and 8) and qualitative comparison of expression patterns from in situ hybridization data provided by the Allen Brain Atlas (*Figure 4c* and *Figure 4—figure supplement 1*; *Lein et al., 2007*). Plotting gene expression with fixed scales across both genes reveals significantly higher expression levels of *Cck* than *Foxb1* in the PMd cluster (column d1 in *Figure 4d*). Simultaneous visualization of both *Cck* and *Foxb1* expression levels within the PMd cluster shows a biased expression pattern of *Cck*- and *Foxb1*-expressing cells towards opposing sides of the PMd cluster (column d2 in *Figure 4d*). There is a substantial number of single-positive cells for each of the two genes (column d3

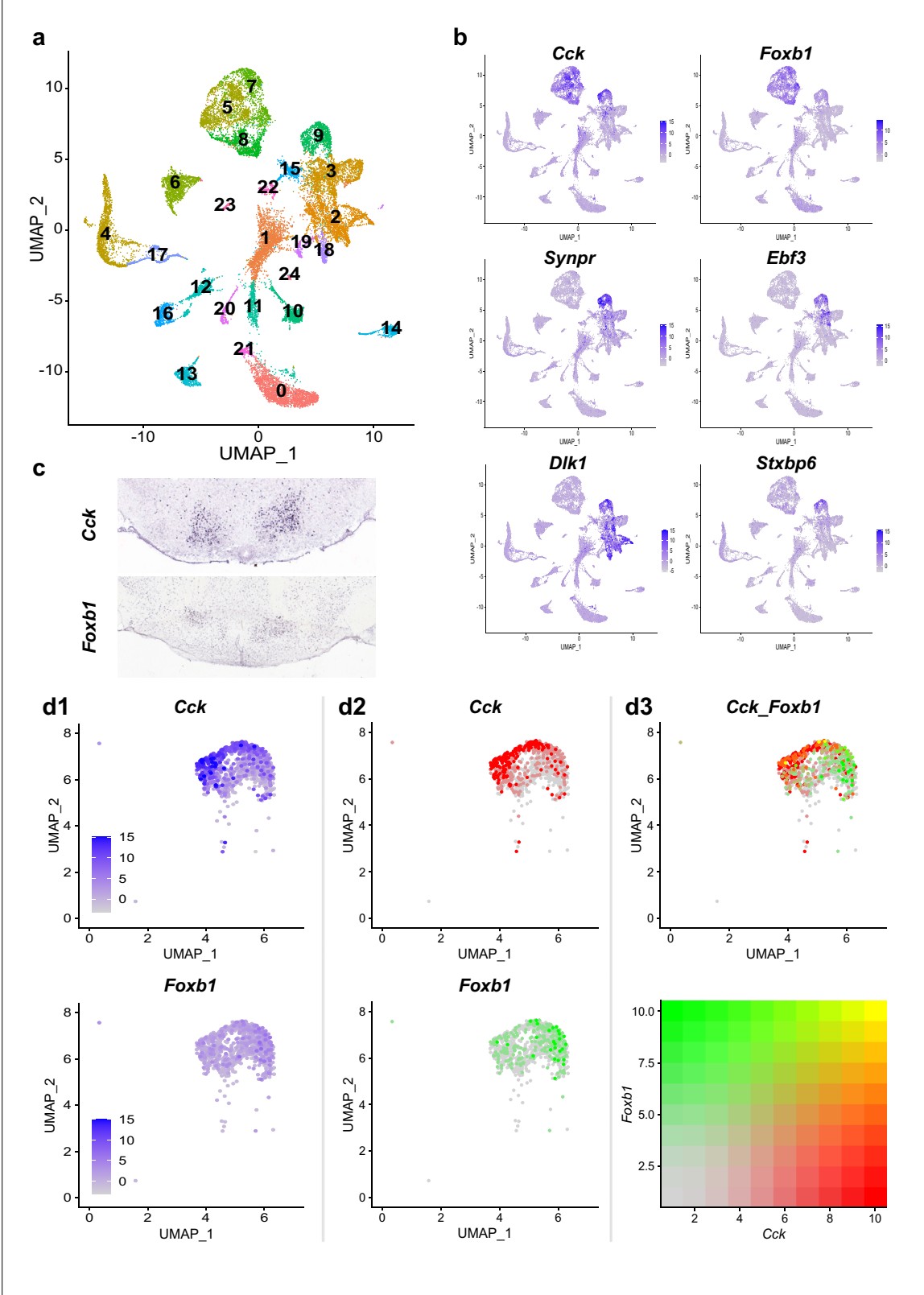

**Figure 4.** Single-cell RNA sequencing reveals differential expression profiles of *Foxb1* and *Cck* in the PMd. (**a**) Single-cell transcriptomic analysis of murine ventral-posterior hypothalamic PMd cells into a distinct cluster (Cluster 9). (**b**) Cluster identity of the PMd cluster is confirmed by expression pattern analysis of genes known to be upregulated in the PMd (Cck, Foxb1, Synpr, Ebf3, Dlk1, and Stxbp6). (**c**) In situ hybridization photomicrographs from the Allen Brain Atlas show the localization of Cck and Foxb1 transcripts in the PMd. (**d**) A magnified UMAP plot representation of the PMd cluster

*Figure 4 continued on next page*

*Figure 4 continued*
highlights the differential expression profiles of Cck and Foxb1 within the PMd cluster. While cells expressing high transcript levels of Cck (middle column, red) preferentially localize to the left side of the PMd cluster, cells with high levels of Foxb1 transcripts (middle column, green) preferentially localize to the opposite (i.e. right) side of the PMd cluster. Analysis of co-expression of Cck and Foxb1 transcripts identifies only few cells as strongly double positive (yellow; see color coding representation), while most cells with high expression levels for one of the two genes have very low to non-existing expression levels of the other gene.

The online version of this article includes the following figure supplement(s) for figure 4:

**Figure supplement 1.** Collection of in-situ hybridization data from the Allen Institute Mouse Brain Atlas for genes used to identify the PMd cell cluster in the scRNA-seq dataset.

in *Figure 4d*). There is indeed a group of PMd neurons co-expressing both markers, but very few of them show high expression levels of both genes.

In summary, although both *Cck* and *Foxb1* are expressed throughout the entire PMd cluster, there are two well-defined, distinct, subpopulations of neurons expressing either *Cck* or *Foxb1* but not both. We were hence able to introduce the presence of the PMd$^{Foxb1}$ as a novel subdivision of the PMd with functional distinction from the PMd$^{Cck}$.

## Hot plate experiments

Two studies had previously identified the parvalbumin$^+$ subpopulation of the parvafox nucleus (parvafox$^{Pvalb}$) to be involved in nociceptive behaviors (*Roccaro-Waldmeyer et al., 2018*; *Siemian et al., 2019*). We therefore extended the scope of our project to further investigate a possible reciprocal effect of the parvafox$^{Foxb1}$ on pain sensation.

In these experiments, we did not observe any significant difference between BL and Stim conditions in any of the three tested animal groups (*Figure 5*, *ChR2-OnTarget_antPAG:* Number of shakes before endpoint $t_{(5)}$ = –0.442, p=0.677 and latency until endpoint $t_{(5)}$ = 0.515, p=0.629; *ChR-OffTarget:* Number of shakes before endpoint $t_{(10)}$ = –1.076, p=0.307 and latency until endpoint $t_{(10)}$ = 1.716, p=0.117; *ArchT3.0-OnTarget_antPAG:* Number of shakes before endpoint $t_{(5)}$ = –0.344, p=0.753 and latency until endpoint $z$=1.150, p=0.875; all reported results are from Welch's t tests, except the last result that was obtained by a Wilcoxon signed-rank test due to non-normality of the data).

These results reveal that the parvafox$^{Foxb1}$ does not excert a reciprocal effect on the parvafox$^{Pvalb}$'s role in pain sensation. Interestingly though, the absence of altered latencies until endpoint behavior

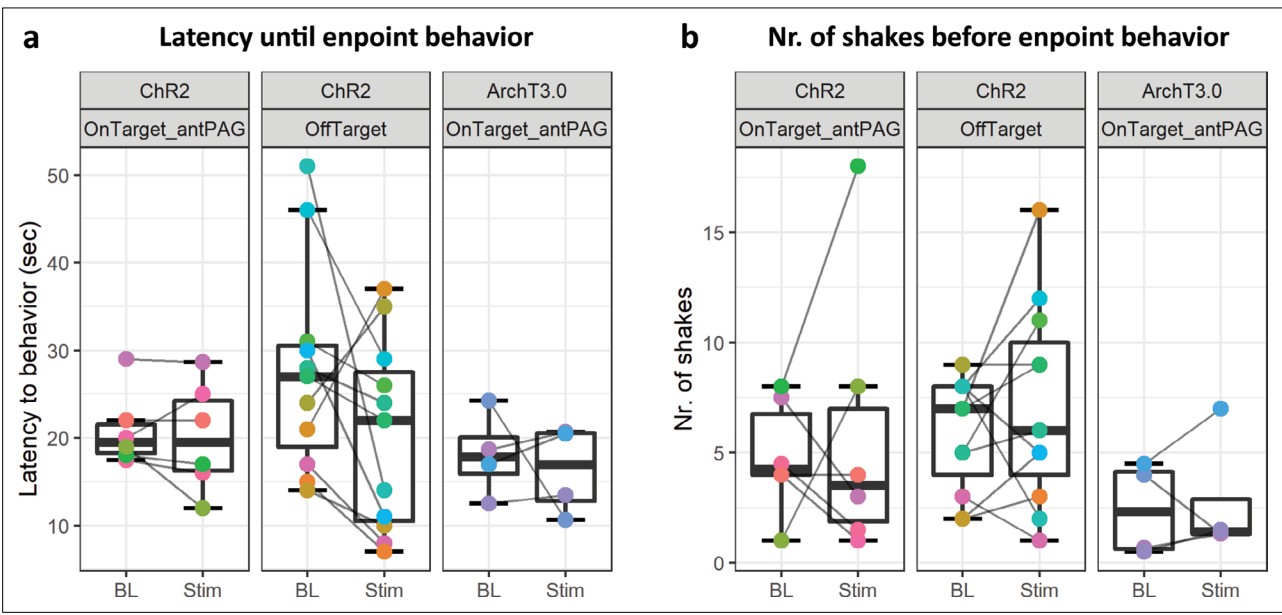

**Figure 5.** Results from hot place experiments performed on Foxb1-Cre animals does not reveal statistically significant differences in (**a**) latency until endpoint behavior (i.e. hindpaw lick(-attempt) or jumping) nor in (**b**) number of shakes before endpoint behavior in any of the three tested groups. These results contradict the hypothesis of a reciprocal effect of the parvafox$^{Pvalb}$ and parvafox$^{Foxb1}$ on pain sensation.

demonstrates that the immobility phenotype of ChR2-expressing *Foxb1*-Cre mice can be escaped, when thermal stimulus intensity reaches nociceptive threshold (*Figure 5a*).

## Alterations of cardiovascular parameters upon optogenetic modulation of *Foxb1* terminals in the dlPAG is observed in a small cohort of animals

The measurement of cardiovascular parameters in a group of 8 *Foxb1*-Cre mice revealed a sudden onset of bradycardia in three mice (106/21–10, 34/21–7, 34/21–10), immediately after starting the optogenetic activation (*Figure 6a*). The heart rate (HR) decreased abruptly by 30–60 beats/min but could reach 290 beats/min in some experiments (*Figure 6b*; 106/21–10). The light-induced cardiovascular responses had an on-kinetic of 1–10 s and at light offset, the HR returned to the baseline levels nearly instantaneously. Augmenting the light intensity of the stimulation increased the extent of HR deceleration. In these three mice, the glass fibers were located bilaterally over the rostral part of the dlPAG (*Figure 7a and b*). Glass fibers located over the Su3 or over the intermediate or caudal dlPAG or the PV2 nucleus did not trigger cardiovascular reactions. In the five *Pvalb*-Cre mice with similar injection of Chr2 in the parvafox$^{Pvalb}$ and glass fibers positioned bilaterally over the rostral dlPAG, no changes in cardiovascular parameters were measured.

In mouse 106-21/10 (*Supplementary file 5*), the HRV changed from 80ms during the baseline period to 120ms during the optogenetic stimulation the STDEV of cycle duration changed from 3.8 ms to 20.6ms [+445%].

The results of these cardiovascular experiments are to be considered as an observational report. Due to a limited number or animals, we cannot draw any statistically valid conclusions at a group level.

## Discussion

Chemogenetic activation of *Foxb1*$^+$ cell bodies in the PMd and parvafox nuclei of the hypothalamus, as well as optogenetic excitation of their axonal endings in the dlPAG both lead to a phenotypic presentation congruent with the reaction observed during innate defensive behavior: tachypnea, immobility, and bradycardia.

As both CNO and clozapine have been reported to activate DREADDs (with different temporal dynamics), we have designed the experimental outline of this study in a way that would allow us to also investigate the effect of both DREADD ligands against each other and against the effect of saline injections. We have shown that injection of both DREADD ligands reproducibly led to the same phenotype in DREADD-expressing animals but not in control animals. The activation of hM3Dq (activating DREADD receptors) expressed in the parvafox$^{Foxb1}$ led to changes in several respiratory parameters, particularly to an increase in respiratory frequency (measured in breaths per minute), consistent with an increased respiratory effort. We infer that this effect is at least partly mediated by the projections of the parvafox$^{Foxb1}$ neurons to the dlPAG. In accordance with this assumption, injections of excitatory aminoacids D,L-hopmocysteic acid (DLH) in the dlPAG raise respiratory activity by an increase in respiratory rate (*Iigaya et al., 2010*; *Subramanian and Holstege, 2014*; *Dampney et al., 2013*).

Immobility is the absence of movements associated with an increase in respiration and muscle tone (*Fanselow, 1994*; *Kalin and Shelton, 1989*) and is consistently correlated with heart rate deceleration (*Vianna and Carrive, 2005*; *Walker and Carrive, 2003*). The body immobility observed in the open field test perturbed the execution and therefore may have influenced the results of the hot-plate test.

In our optogenetic experiments, we cannot differentiate between the respective contributions of the *Foxb1*-terminals belonging to the parvafox$^{Foxb1}$ versus those belonging to the PMd to the behavioral (immobility) and autonomic (bradycardia) effects observed. Are these effects generated by a single common population of neurons, for example, all *Foxb1*$^+$ neurons together (i.e. 'command neurons') or instead by the distinct sub-populations of neurons in the PMd or in the parvafox$^{Foxb1}$?

A subpopulation of neurons in both nuclei express *Foxb1* and a distinct subpopulation consists of parvalbumin positive cells in the parvafox (*Bilella et al., 2014*) and *Cck* positive cells in the PMd (*Wang et al., 2021*). Ten percent of parvalbumin neurons in the parvafox express also *Foxb1*, whereas the situation for *Cck* in the PMd is yet unknown.

The PMd and the parvafox$^{Foxb1}$ originate from the same *Foxb1*$^+$ cell lineage have differential connectivity pattern and as a result they may assume distinct functional tasks (*Bilella et al., 2014*;

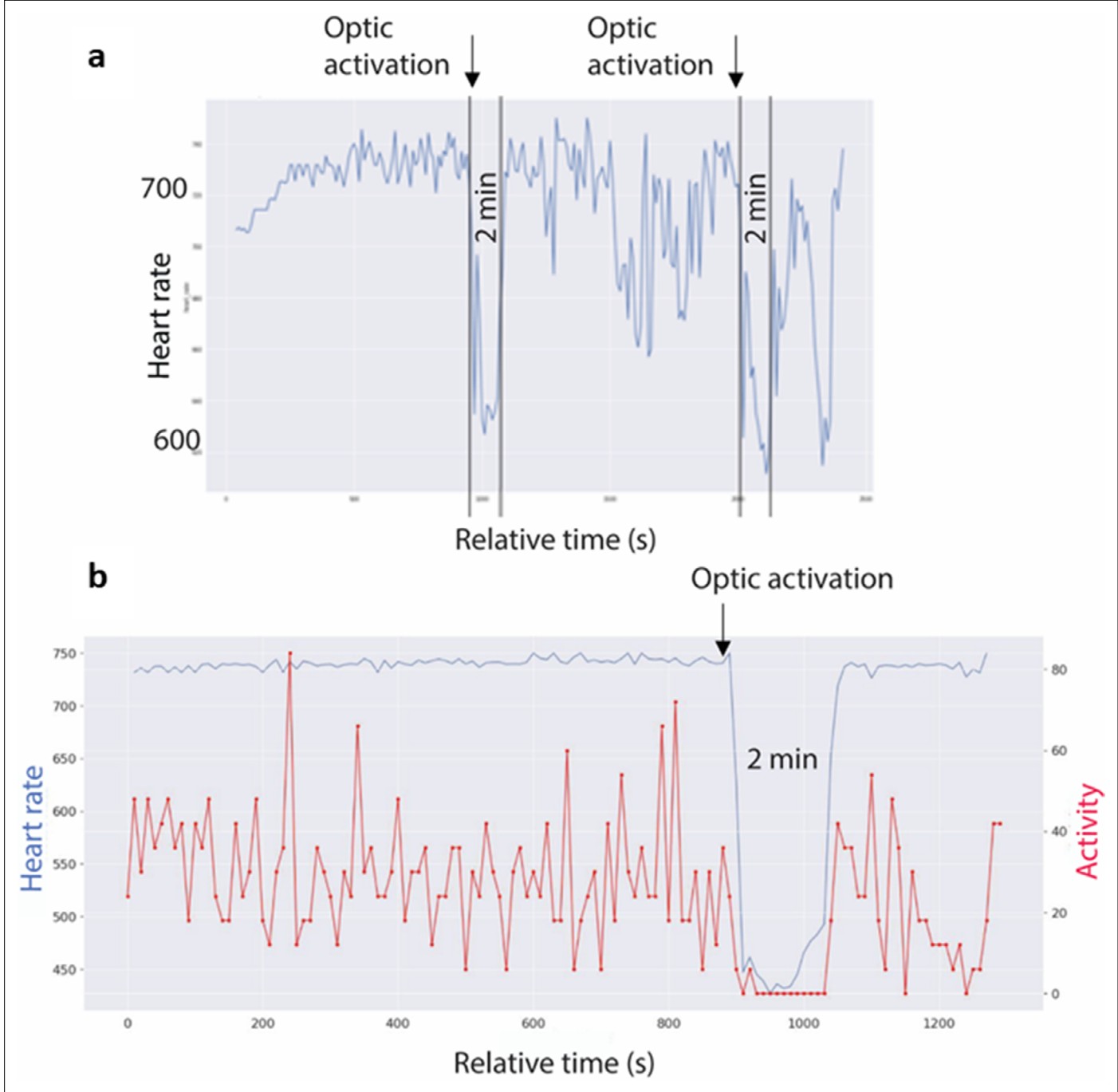

**Figure 6.** Optogenetic activation of Foxb1+ terminals in the rostral dlPAG alters heart rate. (**a**) Telemetric recording of heart rate of mouse 34-21/10 in an open field. Average every 10 s, 40 min long recording. Heart rate increases from 700 to 740 during the first few minutes and remain at this level until the start of the optogenetic activation of the Foxb1 endings in the dlPAG. During the 2 min of flashing blue light, heart rate drops to under 640 but has an initial, short recovery to 690. During the following 10 min of baseline recording, the cardiovascular system is slightly dysregulated, with an increased variability in heart rate. During the second photo-stimulation, the drop in heartbeats has an amplitude comparable to the one during the first stimulation. (**b**) Telemetric recording of heart rate (blue curve) and movements (red curve) of mouse 106-21/10 in an open field. Average every 10 s, 20 min long recording. Heart rate values are around 740 during the first minutes (baseline condition), while the mouse is continuously moving around in the open field. Shortly after the beginning of the optogenetic activation of the Foxb1 endings in the dlPAG, the heartbeat drops massively (by ~40 %) from 740 to 450 bpm and remains around this value during the 2 min of blue light flashing (photo-stimulation). Notice the complete immobility of the mouse during this period. At the end of the optogenetic activation, heart rate rapidly returns to the baseline level of approximately 740 bpm.

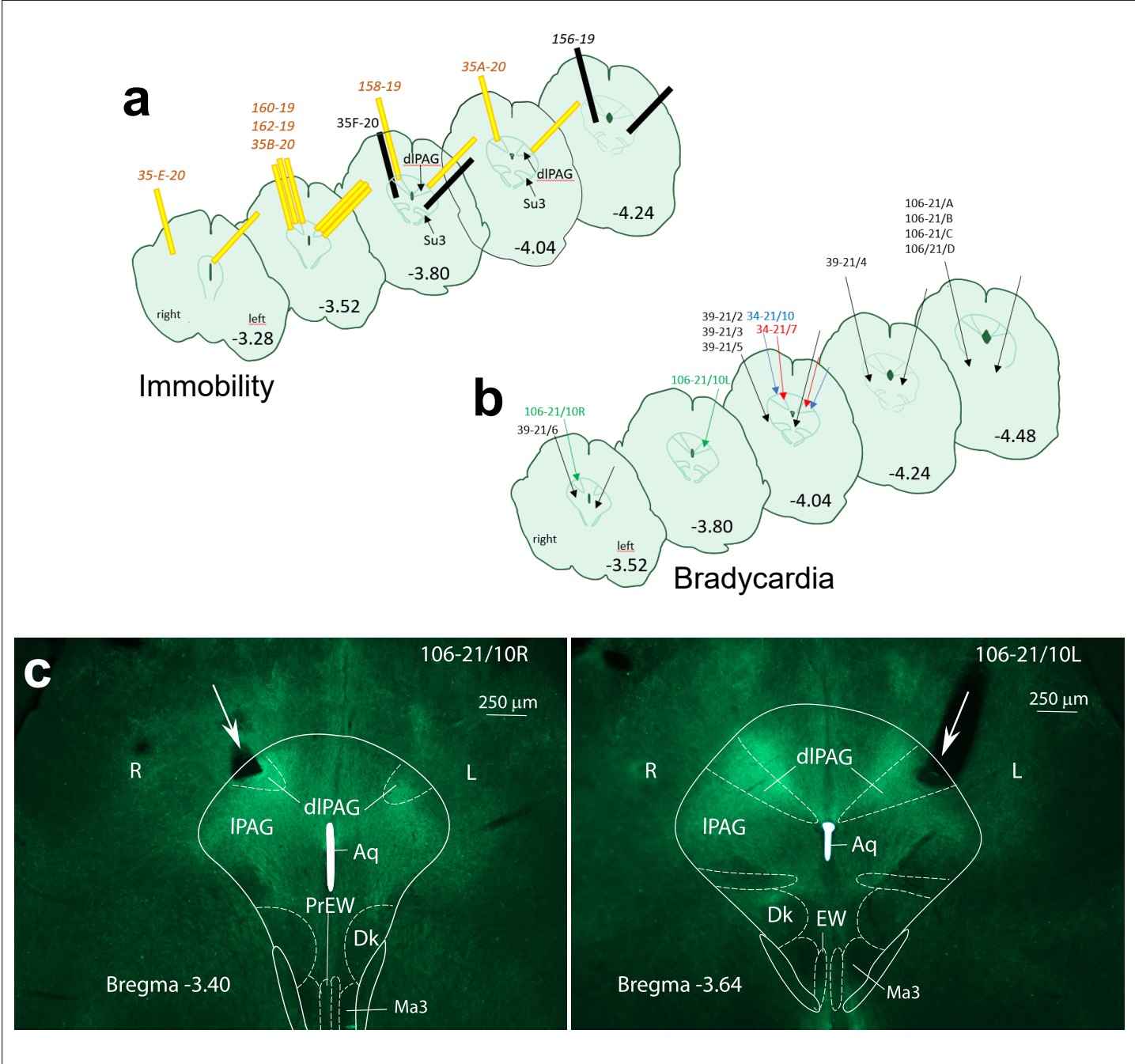

**Figure 7.** Localization of the glass fibers during optogenetic activation of axon terminals in the PAG leading to immobility (**a**), and bradycardia (**b**), respectively. The effects were achieved by inserting the fibers in the rostral part of the dlPAG (Bregma −3.28, −4.04) Glass fibers colored in yellow, with their tip located over the dlPAG (**a**), provoked immobility. Glass fibers colored in black had their tip below the dlPAG and did not evoke changes in mobility. Glass fibers indicated by a green arrow, with their tips over the dlPAG, provoked bradycardia. Glass fibers indicated by black arrows, with their tip below dlPAG or distal to bregma −4.04., did not affect the cardiovascular system (**b**). (**c**) Cross-sections of the rostral PAG with eYFP-labelled Foxb1-terminals located in the dlPAG (mouse 106-21/10). The position of the obliquely inserted glass cannula is indicated with a large white arrow and the flat tip of the cannula is positioned over the dlPAG on the right at bregma −3.40 (left image), and on the left at bregma −3.64 (right image). Aq: Aqueductus cerebri; Dk: Darkschewitsch Nucleus; dlPAG: dorsolateral periaqueductal gray; EW: Edigener-Westphal nucleus; lPAG: lateral PAG; Ma3: medial accessory oculomotor nucleus; PrEW: pre-Edinger-Westphal nucleus.

*Wang et al., 2021*). The PMd receives afferences from the infralimbic and prelimbic areas of the medial frontal cortex (*Comoli et al., 2000*), whereas the parvafox[Foxb1] neurons receives their inputs from the lateral and ventrolateral orbitofrontal cortex (*Babalian et al., 2018*). The neurons of the PMd have a bifurcated output, sending their axons rostrally to the anterior hypothalamic nucleus (AHN) and caudally to the dorsolateral sector of the PAG (*Canteras and Swanson, 1992*). Instead, the efferences of the parvafox[Foxb1] neurons have only minor rostral projections to the septal region and no rostral projections to other hypothalamic nuclei such the SF1[+] neurons of the VHN (*Kunwar et al., 2015*). Instead, the parvafox[Foxb1] rather projects caudally to a wedge-shaped field straddling the lateral (lPAG) and dorsolateral (dlPAG) column (*Bilella et al., 2016*). A further columnar field of terminals is located in the Su3-region of the ventromedial PAG (*Bilella et al., 2016*).

The *Foxb1*[+] neurons of the parvafox nucleus also use glutamate as a neurotransmitter (*Bilella et al., 2016*) and express the gene coding for the neuropeptide adenylate cyclase-activating polypeptide1 (*Adcyap1; Girard et al., 2011*), found to be involved among others in stress disorders (*Ressler et al., 2011*) and cardiorespiratory control (*Shi et al., 2021*; *Barrett et al., 2019*).

The PMd is predestined to play a pivotal functional role in the coordination of defensive behaviors, to both innate and conditioned threats (*Wang et al., 2021*). It consists of a predator activated ventrolateral part (vlPMd) which project to the dlPAG and a dorsomedial part (dmPMd) activated by encounter with aggressive conspecific which avoid projecting to the dlPAG. Ibotenic-acid lesions of the PMd eliminates escape and freezing responses (*Canteras et al., 1997*).

Recording Ca[2+]-transients in PMd[Cck] neurons with fiber photometry, the group of Adhikari (*Wang et al., 2021*), revealed that these neurons are activated during escape but not during freezing and that their chemogenetic inhibition decreases escape speed from threats (*Wang et al., 2021*). The optogenetic activation of the PMd[Cck] cells presumably leads to the activation of the glutamatergic neurons in the dlPAG, which then provoke the fleeing. In our experiments, however, activation of the larger group of *Foxb1*[+] neurons in parvafox[Foxb1] and PMd led to immobility and not to escape. How can this paradoxical effect be explained? One potential explanation would be, that the parvafox[Foxb1] neurons, at least in the rostral dlPAG, innervate a different group of neurons, namely the abundant inhibitory GABA-cells (*Barbaresi, 2005*). It is to be assumed that these short-axon cells, with overlapping distribution in the dlPAG, would locally inhibit the activity of the glutamatergic neurons activated by the PMd[Cck] neurons. Another subgroup of nitric oxide expressing neurons in the dlPAG (*Onstott et al., 1993*) also display a tonic inhibitory effect, via a potentiation of GABAergic synaptic inputs (*Xing et al., 2008*). The role of a fourth group of acetylcholinesterase positive neurons in the dlPAG (*Illing, 1996*) is unknown. The immobility of the mouse would then derive from the local inhibition of the excitatory elements of the dlPAG. Future work will be required to clarify this issue. A second potential explanation is rooted in the expression pattern of *Foxb1* and *Cck* within the PMd. Our reanalysis of the scRNA seq data of the murine ventral-posterior hypothalamus has revealed, that there is a substantial number of *Foxb1* and *Cck* single-positive cells in the PMd. Since our study and the one from Adhikari's group both made use of single-gene Cre knock-in mouse lines, the optogenetic stimulation in both studies led to activation of both single-positive and double-positive neurons. We therefore propose that the observed effects in both studies may be exclusively mediated by single-positive neuronal populations within the PMd. Using an intersectional approach combining Cre- and Flp-recombination dependent expression of optogenetic tools, would allow to investigate whether the same behavioral effects can also be elicited by exclusively activating double-positive (*Foxb1*[+]/*Cck*[+]) neurons within the PMd.

Which other parts of the brain are recruited by the dlPAG neurons to initiate bradycardia and immobility? The dlPAG projects to the cuneiform nucleus (*Meller and Dennis, 1991*), which is involved in defensive locomotion (*Jordan, 1998*) through its connection with the lateral gigantocellular reticular nuclei (*Tovote et al., 2016*), implicated in high-speed locomotion (*Caggiano et al., 2018*). Additional targets of the cuneiform projections are the motor nucleus of the vagus and the nucleus tractus solitarius (*Korte et al., 1992*) and a few immunostained boutons were found around the nucleus ambiguus in the rostroventrolateral medullary nucleus (Figure 5K in *Korte et al., 1992*). These projections are probably involved in the bradycardic response to the optogenetic stimulation in the dlPAG. The projections of the cuneiform nucleus to the rostral ventrolateral medulla promote sympathetic vasomotor activity (*Verberne, 1995*). An alternative pathway for the expression of the behavioral and autonomic effects of dlPAG activation is through a link with the dorsomedial hypothalamus (DMH; *Dampney et al., 2013*).

Bradycardia starts immediately after the beginning of the optogenetic activation of the *Foxb1*[+] terminals in the dlPAG. The increase in the HRV during the optogenetic stimulation indicates an intervention of the parasympathetic system on the sinoatrial node of the heart (*Standish et al., 1995*). The parasympathetic impact on the heart is mediated by acetylcholine neurotransmission (*Wu et al., 2019*) and has a very short latency of response, with peak effect at about 0.5 s and return to baseline within 1 s (*Pumprla et al., 2002*).

During ultrasonic vocalization tests in our laboratory (data not reported in this study), we observed that the *Foxb1*-Cre mice did not escape the ChR2-induced state of immobility, even when another genotype-matched and novel intruder mouse was placed into the ChR2-expressing mouse's home cage. When approached by the intruder mouse, the mice were still able to move their heads under photo-stimulation, did sniff and interacted with the intruder mouse but did not actively follow the intruder. A mouse displaying complete immobility remained completely motionless even when the intruder was actively approaching it and even when it started digging into the cage bedding underneath the resident mouse.

In an interesting recent publication of Chen's group (*Liu et al., 2022*), the projection of the parvafox*Foxb1* to the lateral PAG (lPAG) indeed has been found to drive social avoidance in mice. Inhibition of the parvafox*Foxb1* neurons by the GABAergic input from the lateral septum reversed the deficit in social novelty preference induced by chronic social defeat stress. The authors conclude that activation of the parvafox*Foxb1* leads to avoidance of general social situations.

Hence, our observations of a reduced social interaction phenotype agree with the results from Chen's group. However, since activation of the parvafox*Foxb1* terminals in the dlPAG also leads to immobility of the mouse in the absence of any social stimulus, we argue that the outcome of the social avoidance test could have been influenced by the failing motor performance of the *Foxb1*-Cre mice and might not be of pure social nature. In light of these observations, it is further interesting to note, that the results from our hot plate experiments indicate that the immobile/hyporeactive phenotype of these mice can indeed be escaped, when an adequate stimulus is presented (e.g. thermal stimulus intensity reaching nociceptive levels).

The possibility of antidromic propagation of ChR2-mediated action potentials poses a major caveat to the interpretation of results obtained from experiments in which presynaptic terminals are optogenetically activated. This mechanism could lead to coincident activation and realease of neurotransmitters from collaterals originally not intended to be targeted by the intervention. The use of inhibitory optogenetic tools, however, circumvents such a mechanism of uncontrolled backpropagating action potentials (*Rost et al., 2022*). In our optogenetic experiments in which we have used the inhibitory opsin ArchT3.0, we detected statistically significant differences to the baseline conditions in multiple locomotive parameters. This observation of bidirectional modulation is an indication of at least some baseline activity of the *Foxb1*[+] neurons in the hypothalamus in our experimental settings. Importantly, these results and observations additionally support the notion that the locomotor phenotype is due to the specific projection of *Foxb1*[+] neurons onto the PAG. If the hypolocomotor phenotype in ChR2 expressing mice was driven by backpropagation-mediated activation of collaterals, the presynaptic ArchT3.0-mediated inhibition of the *Foxb1*[+] projections to the PAG would be unlikely to cause an effect opposing the ChR2-mediated phenotype. In our ArchT3.0 experiments, we specifically employed a stimulation paradigm in which we had light illumination on for 10 s and then off for another 10 s. This was done to reduce pH-mediated side effects of ArchT3.0 axonal activation observed with sustained stimulation, like spontaneous release of neurotransmitters (*Mahn et al., 2016*). If such side effects had occured in our experiments, we would expect to see a phenotype that resembles the one observed in the ChR2-mediated activation. However, in our experiments, we saw no indication of such a behavior. Recently, two new optogenetic tools have been developed that allow efficient and targeted presynaptic silencing through GPCRs without the alteration in intracellular pH levels (*Mahn et al., 2021*). For future experiments, such tools might be more suitable for presyaptic silencing experiments.

In summary, our results demonstrate evidence for a role of the hypothalamic *Foxb1*[+] neuronal population located in the parvafox*Foxb1*, and the PMd in the expression of bradycardia and immobility, as well as in increasing respiratory rate. This effect is in accordance with distinctive coping patterns, the freezing-like behavior accompanied by bradycardia being associated with innate defensive behavior.

In view of the recent finding that PMd neurons are activated during escape (*Wang et al., 2021*), the immobility that we observe could be mediated by the terminal endings of the parvafox*Foxb1*, synapsing

on the inhibitory GABA-interneurons of the rostral third of the dlPAG or could be the result of the activation of the *Foxb1* single-positive PMd neuronal population (i.e. *Foxb1*⁺/*Cck*). Further investigations on the neuronal activity of these hypothalamic *Foxb1*⁺ populations during exposure to natural threats will allow for a better understanding of the differential roles of *Foxb1*⁺ and *Cck*⁺ PMd neurons. Such experiments could for example consist of fiber photometry calcium recordings or microendoscopic calcium imaging experiments during predator exposure in *Foxb1* and/or *Cck* Cre-driver lines.

# Materials and methods

**Key resources table**

| Reagent type (species) or resource | Designation | Source or reference | Identifiers | Additional information |
|---|---|---|---|---|
| Gene (*Mus musculus*) | *Foxb1* | NCBI Gene | Gene ID: 64290 | |
| Genetic reagent (*M. musculus*) | Foxb1^tm1(cre)Gabo | MGI | MGI ID: 3772366 | |
| Genetic reagent (*M. musculus*) | 129P2-Pvalb^tm1(cre)Arbr/J | MGI | MGI ID: 3773708 | |
| Genetic reagent (adeno-associated virus) | AAV5-EF1α-DIO-hChR2(H134R)-eYFP | Addgene | Catalog #: 20298-AAV5 | |
| Genetic reagent (adeno-associated virus) | AAV2-EF1α-DIO-eArchT3.0-eYFP | University of North Carolina, Vector Core | | Kind gift from Adamantidis Lab |
| Genetic reagent (adeno-associated virus) | AAV2-hSyn-DIO-hM3D(Gq)-mCherry | Addgene | Catalog #: 44361-AAV2 | |
| Genetic reagent (adeno-associated virus) | AAV2-hSyn-DIO-hM4D(Gi)-mCherry | Addgene | Catalog #: 44362-AAV2 | |
| Antibody | anti-mCherry (Rabbit polyclonal) | Abcam plc. | product code: ab167453 | IF(1:1000) |
| Antibody | anti-c-Fos (Mouse monoclonal) | Abcam plc. | product code: ab208942 | IF(1:2000) |
| Antibody | anti-GFP (Chicken polyclonal) | Aves Labs, Inc. | Product code: GFP-1020 | IF(1:200) |
| Antibody | anti-mouse IgG (Horse polyclonal, biotinylated) | Vector Laboratories | Product code: BA-2000 | IF(1:200) |
| Antibody | anti-chicken IgG (Donkey polyclonal, Cy2-conjugated) | Jackson ImmunoResearch | Product code: 703-225-155 | IF(1:200) |
| Antibody | anti-Rabbit IgG (Donkey polyclonal, Cy3-conjugated) | Jackson ImmunoResearch | Product code: 711-165-152 | IF(1:200) |
| Peptide, recombinant protein | Streptavidin (Alexa Fluor 647-conjugated) | Jackson ImmunoResearch | Product code: 016-600-084 | IF(1:200) |
| Sequence-based reagent | EGFP-f | This paper | PCR primer | CTC GGC ATG GAC GAG CTG TAC AAG |
| Sequence-based reagent | GAB20 | This paper | PCR primer | CAC TGG GAT GGC GGG CAA CGT CTG |
| Sequence-based reagent | GAB22 | This paper | PCR primer | CAT CGC TAG GGA GTA CAA GAT GCC |
| Chemical compound, drug | DAPI | Life Technologies Corporation | Product code: D1306 | IF(1:5000) |
| Chemical compound, drug | Clozapine | Sigma-Aldrich | Product code: C6305 | |
| Chemical compound, drug | CNO | Sigma-Aldrich | Product code: SML2304 | |
| Software, algorithm | Ponemah Software | DSI | PNM-P3P-CFG | |
| Other | Whole-body barometric plethysmography chambers | DSI | Product code: 601-0001-011 | Used in respiration-related experiments |
| Other | Buxco Bias flow pump | DSI | Product code: 601-2201-001 | Used in respiration-related experiments |

*Continued on next page*

*Continued*

| Reagent type (species) or resource | Designation | Source or reference | Identifiers | Additional information |
|---|---|---|---|---|
| Other | Buxco differential pressure transducer | DSI | Product code: 600-1114-002 | Used in respiration-related experiments |
| Other | Temperature and humidity probe | DSI | Product code: 600-2249-001 | Used in respiration-related experiments |
| Other | ACQ-7700 USB amplifier | DSI | Product code: PNM-P3P-7002SX | Used in respiration-related experiments |
| Other | Dual LED light source | Prizmatix Ltd. | Product code: 34117 | Used for optogenetic experiments. blue (peak $\lambda$ =453 nm) and lime green (peak $\lambda$ =536 nm) |
| Other | 1.5 mm optical fiber | Prizmatix Ltd. | Product code: 34131 | Used for optogenetic experiments. Connecting LED light source to rotary joint |
| Other | Rotary joint | Prizmatix Ltd. | Product code: 43043 | Used for optogenetic experiments. |
| Other | Dual fiber patch cord | Prizmatix Ltd. | Product code: 34115 | Used for optogenetic experiments. 2x500 µm diameter |
| Other | Ceramic cannulas | Prizmatix Ltd. | Product code: 43071 | Used for optogenetic experiments. 1.25 mm outer diameter; 230 µm inner diameter |
| Other | 0.66 NA optical fibers | Prizmatix Ltd. | | Used for optogenetic experiments. 200 µm outer diameter. |
| Other | 473 nm DPSS laser | Laserglow technologies | Product code: LRS-0473-PFO-00500–01 | Used for optogenetic experiments. |

## Mice

A total of 42 mice of both sexes were used for the purpose of this study. Most animals were Foxb1[tm1(cre)Gabo] mice that express Cre-recombinase under the control of the promoter for *Foxb1*. For the cardiovascular experiments, five mice belonging to the *Pvalb*-Cre strain (129P2-Pvalb[tm1(cre)Arbr]/J) were used (*Hippenmeyer et al., 2005*). All animals were maintained at a constant temperature of 24 °C in state-of-the-art animal facilities with a 12hr-light/12hr-dark cycle and had ad libitum access to food and water.

All experiments were approved by the Swiss federal and cantonal committee for animal experimentation (2016_20E_FR and 2021–10-FR) and were conducted in accordance with the institutional guidelines of the University of Fribourg.

## Intracerebral AAV injection

Mice in which we aimed at modulating neuronal activity were injected intracerebrally either with (i) Channelrhodopsin (AAV5-EF1α-DIO-hChR2(H134R)-eYFP), (ii) Archeorhodopsin (AAV2-EF1a-DIO-eArchT3.0-eYFP), (iii) activating DREADD (AAV2-hSyn-DIO-hM3D(Gq)-mCherry) or, (iv) inhibiting DREADD (AAV2-hSyn-DIO-hM4D(Gi)-mCherry). All optogenetic and chemogenetic agents injected into *Foxb1*-Cre and *Pvalb*-Cre mice were Cre-dependent and were bilaterally injected into the posterior portion of the parvafox nuclei of the LHA.

All intracerebral viral vector injections were conducted according to the following standard protocol:

The animal was weighed and was anesthetized with an intraperitoneal injection (i.p.) of a mixture of ketamine (40–60 mg.kg$^{-1}$ of body weight) and xylazine (10–15 mg.kg$^{-1}$ of body weight) diluted in physiological (0.9 %) saline. The fur covering the cranium was shaved and the mouse was head fixed into a stereotaxic apparatus (Kopf instruments, model 5000) equipped with a heating pad to maintain

body temperature. Depth of anesthesia was assessed regularly throughout the entire duration of the surgery. If necessary, additional doses of the anesthetic agent were injected. Eye ointment was applied, and the eyes were protected from exposure to direct light. Once tail pinch and toe pinch reflexes were vanished, a sagittal skin incision above the midline of the cranium was performed and the cranial sutures were identified to locate bregma. Craniotomy was performed bilaterally above the site of injection with a dental steel bur. The viral vector was then aspirated into a 2.5 µl Hamilton syringe via a fine-bored 34-gauge needle (external diameter of 0.14 mm). The Hamilton syringe was mounted onto a manual microinjection unit and fixed to the stereotaxic frame. After identification of bregma, the needle tip was placed just above it and the anterior-posterior and medial-lateral coordinates for the injection sites were calculated relative to bregma (AP –1.3 mm, ML +/-1.3 mm). The needle was then placed on the brain surface above the target injection site and the dorsal-ventral coordinate was calculated relative to the brain surface (DV –5.5 mm). The needle was subsequently lowered into the brain until the desired depth was reached. 200 nl of the viral construct were injected into the parvafox nuclei bilaterally at a rate of 100 nl/min. Before retraction of the needle, the needle was left in place for 5 min to allow diffusion of the virus and to minimize backflush of the virus along the entry path of the needle. The same procedure was then repeated on the contralateral side. Once both viral injections were completed, the mouse was released from the stereotaxic frame and the skin incision was closed with one to two surgical stiches. The mouse was placed into a separate cage to recover from the surgery and was later put back into its home cage once consciousness was regained.

## Fiber optic cannulas and implantation procedure

Fiber optic cannulas were custom made with ceramic ferrules (230 µm ID, 1.25 mm OD; Prizmatix Ltd., Israel) and optical fibers (200 µm OD, 0.66 NA; Prizmatix Ltd., Israel). The length of the protruding glass fiber end was set to 0.4 mm for the cannulas to be implanted above the dlPAG. The non-protruding end of the optical fiber was polished on a series of lapping sheets with decreasing grit size (5 µm, 3 µm, 1 µm, 0.3 µm). Each optical fiber implant was measured for its coupling efficiency before implantation. Implantation of the fiber optic cannulas was performed 2–3 weeks after virus injection.

The surgical procedure for implantation of the cannulas was as follows:

Initial preparation of the animal for access to the skull was performed as described above for viral vector injections. Once bregma was identified, small marks on the skull were made to identify the position of the bilateral bur holes for the cannulas and the three to four skull fixation screws. The bur holes were hand drilled with a dental steel bur and three to four fixation screws were screwed into the skull. The skull fixation screws were additionally locked to the skull by cyanoacrylate adhesive. A cannula was then mounted to the stereotaxic frame via a standard electrode holder (Kopf instruments) and was inserted into the brain at the desired coordinates. The bilateral cannulas implanted above the dlPAG were inserted at a 20° angle in the coronal plane (V-shaped arrangement of cannulas) to allow enough space for later connection to the patch cords. The coordinates for dlPAG cannulas insertion sites were AP –4.0 mm, ML +/-1.5 mm relative to bregma and insertion depth (at an angle of 20°) was –2.8 mm from the surface of the skull. Once the fibers were in place, self-curing acrylic (Palidur powder and liquid, Heraeus Kulzer GmbH, Hanau, Germany) was applied to the skull, the ceramic ferrule and the fixation screws. The mouse was then released from the stereotaxic apparatus and a 0.3 ml subcutaneous injection of physiological (0.9 %) saline into the neck scruff was made to support recovery from the surgery. Further, Carprofen (Rimadyl, 5 mg.kg⁻¹ of body weight) was subcutaneously administered as an analgesic following surgery. After surgery, the mouse was placed in a separate cage under a warming lamp until consciousness was fully regained. Whenever possible, mice were put back into their home cages together with their cage mates. If fighting behavior was observed between male cage mates, the mice were kept in separate cages.

## Optogenetic stimulation

Time-resolved optogenetic gain-of-function manipulation of *Foxb1*[+] axon terminals was performed bilaterally. A dual LED light source (MP-Nr: 34117; Prizmatix Ltd., Israel) with blue (peak $\lambda$ =453 nm) and lime green (peak $\lambda$ =536 nm) light emitting diodes (LEDs) were connected to a 1.5 mm optical fiber (NA 0.63; MP-Nr: 34131; Prizmatix Ltd.) which terminated into a rotary joint (MP-Nr: 34043, Prizmatix Ltd.), connecting to a dual fiber patch cord (2x500 µm, NA 0.63, MP-Nr: 34115; Prizmatix Ltd.). The two patch cord ends were attached to the cannulas on the head of the mouse by ceramic

sleeves (ID = 1.25 mm, MP-Nr: 34071; Prizmatix Ltd.). ChR2 mice were stimulated with an intensity of 7–15 mW per fiber tip. The protocol for ChR2 activation consisted of bursts of 500ms duration with an intraburst frequency of 30 Hz, a pulse duration of 5ms and an interburst interval of 500ms and was based on the previously published firing properties of *Foxb1*$^+$ neurons of the medial mammillary complex (*Alonso and Llinás, 1992*). ArchT3.0 stimulation protocol consisted of alternating 10 s windows with continuous LEDon and LEDoff, respectively (intensity of 7–15 mW). The optogenetic pulses were generated on a PulserPlus (MP-Nr.:34192, Prizmatix Ltd.) installed on a PC and were sent to the dual LED source and to an ACQ-7700 USB amplifier (PNM-P3P-7002SX, Data Sciences International [DSI], St. Paul, MN, USA).

A laser system of the type LRS-0473-PFO-00500–01 LabSpec (473 nm DPSS) from Laserglow technologies, North York, Ontario, Canada, was employed for augmenting the power output in the cardiovascular experiments (70–222 mW). The estimated power at the specimen was measured with a photodiode (Thorlabs).

## Chemogenetic stimulation

Mice stereotaxically injected with activating or inhibiting DREADDs, as well as animals without expression of any DREADDs (i.e. DREADD_neg) were injected i.p. with CNO, clozapine or physiological saline in a given experimental block. CNO was administered at a dose of 1 mg·kg$^{-1}$ of bodyweight 30 min before the start of the experiment. Due to faster pharmacokinetics, clozapine was administered i.p. at a subthreshold dose of 0.1 mg·kg$^{-1}$ of bodyweight immediately before the start of the experiment. Attention was given to eventual backflush or unintentional subcutaneous administration. Dose calculations were adjusted individually to bodyweight before each injection.

## Whole body barometric plethysmography (WBP)

To measure respiratory parameters, mice were placed in whole-body barometric plethysmography chambers (item Nr. 601-0001-011, DSI) that were connected to a Buxco Bias flow pump (item Nr. 601-2201-001, DSI, St. Paul, MN, USA) to avoid $CO_2$ accumulation inside the chambers and to circulate ambient air through the chambers at a rate of 1 L·min$^{-1}$. Each chamber was equipped with a high sensitivity differential pressure transducer (Buxco TRD5700 Pressure, item Nr. 600-1114-002, DSI) as well as a temperature and humidity probe (item Nr. 600-2249-001, DSI). All sensors were connected to the ACQ-7700 amplifier for pre-processing of signals.

Acquisition and processing of respiratory signals as well as the TTL signal from the LED pulser was performed within the Ponemah Software (PNM-P3P-CFG, DSI). The respiratory flow signals were sampled at a rate of 500 Hz and were filtered with a 30 Hz low pass filter. Using the Unrestraint Plethysmography Analysis Module (URPM) within Ponemah, respiratory cycles were validated in the flow signal and several respiratory parameters were derived from them. For each animal, the attribute analysis settings were individually adjusted to obtain optimal validation marks of respiratory cycles. The derived respiratory parameters were then averaged into 5 s averages and exported as excel files for further statistical analysis and plotting in R and RStudio (RStudio, Inc, Boston, MA, USA).

On the experimental day, mice were brought into the experimental room and were allowed to habituate to the new environment for at least 45 min. Each animal had two habituation sessions of 90 min on two different days prior to the first baseline measurements. For these habituation sessions, DREADD mice were injected with 0.2 ml of physiological (0.9 %) saline i.p. 30 min before they were placed into the WBP chamber.

Each animal's baseline (BL) was measured for 3x90 min on 3 consecutive days. BL condition for DREADD animals consisted in a 0.2 ml i.p. injection of physiological (0.9 %) saline 30 min before measurement. After BL recordings, DREADD animals were measured for 3x90 min under CNO stimulation on 3 consecutive days and for another 3x90 min under clozapine stimulation on another 3 consecutive days. Mice were given a break of at least 3 days between CNO and clozapine experimental blocks to allow for complete clearance of the substances and to reduce stress on the animal. WBP chambers were cleaned with soap and water after each recording to avoid olfactory stimulation of the next mouse. All WBP experiments for a given mouse were performed at the same time of the day to account for circadian variability.

## Cardiovascular measurements with telemetry

For measuring cardiovascular functions, an implantable telemetry system from Datascience International (DSI) was used. The transmitter was implanted in 13 mice following manufacturer instructions and as previously described (*Huetteman and Bogie, 2009*; *Pillai et al., 2018*). Briefly, mice were anesthetized with the aid of isoflurane and implanted with a PA-C-10 transmitter (DSI). This small pressure sensing telemetry tool was implanted in the left carotid artery taking care to place the pressure-sensitive tip in the aortic arch. The radio transmitting device (RTD) was placed under the skin along the right flank of the mice. Mice were given analgesics for 3 days post-surgery. One week after telemetric sensor implantation, a test measurement was conducted to ensure that the catheter, the RTD and the receiver (RPC-1) were functioning well. The optogenetic experiment was performed after 2 more weeks.

Measurements were recorded with Dataquest ART (version 3.1) and RespiRate (DSI). Recordings were taken three times for 3 weeks post-surgery. Recordings were continuously taken for 30 s every 5 min during these sessions. Systolic, diastolic and mean blood pressure, heart rate and activity were analyzed. Pulse wave signals were used to measure the heart rate variability (HRV).

## Open field test

As in all other experiments reported in this paper, mice were transported into the experimental room and were allowed to acclimatize to the new environment for at least 45 min. The experimental arena for the open field test consisted in a 40x40 cm cage with transparent plexiglas walls and a gray colored metal floor. After each recording of a mouse, the arena was thoroughly cleaned with 70% ethanol. Each mouse was recorded four times (2 x saline/LEDoff and 2 x clozapine/LEDon) spread across 2 days with at least 2 days between experimental days. In DREADD experiments, the saline condition was performed in the morning and the clozapine session in the afternoon. To account for potential circadian bias, optogenetic morning and afternoon sessions for each mouse were alternated, so that the two recordings for LEDoff and LEDon, respectively, were recorded once in the morning session and once in the afternoon session. Recordings consisted of a 5-min habituation period inside the experimental arena, uninterruptedly followed by a 5-min recording period. Mice for DREADD experiments were injected with clozapine (i.p. 0.1 mg·kg⁻¹ of bodyweight) 30 min before the recording time window. Optogenetic mice remained in their home cages until the start of the 5-min habituation session, shortly before which they were attached to the patch cords. In all optogenetic animals (including controls), a stop watch signal marked the end of the 5 min of habituation. The experimenter then initiated the LEDon period accompanied by another brief auditory signal.

## Pose estimation of open field data

For body part tracking, DeepLabCut (version 2.2.2) (*Mathis et al., 2018*; *Nath et al., 2019*) was used. Specifically, 500 frames taken from 25 videos were labeled and 95% was used for training. A ResNet-50-based neural network with default parameters was used for 600,000 training iterations. We validated with 1 shuffle, and found the test error was: 2.03 pixels, train: 2.39 pixels (image size was 1280 by 720 pixels). We then used a p-cutoff of 0.95 to condition the X,Y coordinates for future analysis. This network was then used to analyze videos from similar experimental settings.

## Hot plate test

To assess thermal nociceptive perception in optogenetic *Foxb1*-Cre mice, they were first connected to the patch cords and placed onto an insulation layer of cork and several layers of paper towels on the hot plate for 3 min to acclimatize to the new environment. The hot plate (Analgesia meter for rodents, IITC life sciences Inc, Woodland Hills, CA, USA) was maintained at 51+/-0.1 °C. After 3 min had passed, the insulation layer was removed and the latencies until hindlimb shaking, hindlimb licking and jumping were recorded as baseline. In case the mouse did not display any of the two endpoint behaviors (i.e. hind paw licking or jumping), the recording was terminated 50 s after the onset of the thermal stimulus to prevent tissue damage. After termination of the baseline recording, the insulation layer was again placed between the mouse and the hot plate and the optogenetic stimulation was initiated (see 'Optogenetic stimulation' for detailed parameters). After 3 min of optogenetic stimulation, the insulation was once again removed and the mouse was placed back onto the hot plate, while the stimulation continued. Just like in the baseline condition, the same latencies were recorded, or the

experiment was terminated after maximally 50 s. As observed in pilot experiments, the ChR2-injected mice displayed reduced locomotor activity and seemed to be limited in their ability to lick their hind paw. We therefore also terminated the recording, when the mouse displayed an obvious attempt to lick its hind paw and the latency until the display of such an event was recorded. The same procedure was repeated one more time on another day to record two baseline and two LED condition recordings for each optogenetic animal.

## Histology

Before mice were perfused, they went through the same routine as in a regular experimental condition (LED stimulation or CNO injection) for later detection of c-Fos immunofluorescence. Two hours after the start of the activation/inhibition experiments, the mice were deeply anesthetized with the same anesthetic agent used for surgical procedures (see above). Once the pain reflexes vanished, the thorax was fenestrated, and the mouse was transcardially perfused with physiological (0.9 %) saline for 3 mins and subsequently with 4% paraformaldehyde (PFA) in PBS 0.1 M (pH 7.4) for 5 min. Decapitation was performed and the head was placed in 4% PFA until extraction of the glass fiber implants. After cannulas extraction from the skull, the cannulas were stored for post-extraction coupling efficiency measurement. The brains were then harvested and further immersed in TBS 0.1 M+18% sucrose +0.02% Na-azide overnight for cryoprotection.

Brain tissue was cryo-sectioned into 40-μm-thick sections on a sliding microtome (SM 2010R, Leica) connected to a freezing unit (Microm KS 34, Thermo Fisher Scientific) and every sixth section was selected for incubation with antibodies.

The histological processing of free-floating sections was performed as follows:

Sections were washed for 3x5 min in TBS 0.1 M. The sections were then incubated for 2 days at 4 °C with the primary antibodies at their corresponding dilutions (see *Supplementary file 6*) in TBS 0.1 M+0.1% Triton X-100 and 10% bovine serum (BS). After incubation with the primary antibodies, sections were washed for 3x5 min in TBS 0.1 M and further incubated for 2 hr at room temperature with a biotinylated antibody diluted 1:200 in TBS 0.1 M and 10% BS. Subsequently, sections were washed 1x5 min in TBS 0.1 M and 2x5 min in Tris pH 8.2 before they were incubated with Cy2-, Cy3- and Cy5-conjuncated secondary antibodies or streptavidin, which were diluted 1:200 in Tris pH 8.2 for 2 hr at room temperature. After incubation with Cy-conjugated secondary antibodies and Cy-conjugated streptavidin, the sections were washed 1x5 min in Tris pH 8.2 and 2x5 min in TBS 0.1 M and then incubated with DAPI 1:5000 in TBS 0.1 M for 5 min at room temperature.

Sections were mounted onto Superfrost +glass slides (Thermo Scientific) and were left to dry for 2 hr at 37 °C. Slides were then quickly washed in dH$_2$O before standard cover slips were mounted to the slides with Hydromount mounting medium (National Diagnostics, Atlanta, GA, USA).

## Genotyping

Genotyping was performed before animal selection and again after perfusion to exclude any mix-up during the testing period. For the first genotyping, tissue samples from the toe clipping procedure were used. For the second genotyping, tissue samples were taken from the tail after the animal was deeply anesthetized and before perfusion with 4% PFA.

The primer sequences used for *Foxb1*-Cre genotyping were as follows:

-EGFP-f: 5'-CTC GGC ATG GAC GAG CTG TAC AAG-3'
-GAB20: 5'-CAC TGG GAT GGC GGG CAA CGT CTG-3'
-GAB22: 5'-CAT CGC TAG GGA GTA CAA GAT GCC-3'

## Reanalysis of single-cell RNA sequencing data set

The raw data used for the reanalysis of single-cell RNA sequencing reads from mouse (postnatal day 30–34) ventral-posterior hypothalami was kindly made available to the public by the original authors (*Mickelsen et al., 2020*) through the gene expression omnibus under the accession number 'GSE146692'.

The entire analysis workflow was executed as follows using the Seurat package V4.1.1 in R (*Hao et al., 2021*):

Datasets of two male and two female animals were imported into R and initialized as four separate Seurat objects. Subsequently, the four separate objects were merged into a common Seurat object and the percentage of mitochondrial transcripts as well as the number of hemoglobin transcripts per cell were calculated. Quality control was performed by analyzing number of features, number of counts, percentage of mitochondrial RNA, and number of hemoglobin gene transcripts. Data meeting the following criteria were kept for downstream analysis: nFeatures >200 & nFeatures <7500 & percent.mtRNA <15 & nHemoglobin_RNA <50. The filtered dataset was then normalized within each original identity (male1, male2, female1, female2) using the SCTransform function and 3000 integration features were selected. To correct for batch effects, integration anchors were detected based on the selected integration features prior to integration of the four datasets. Next, uniform manifold approximation and projection (UMAP) was performed for dimensional reduction before clusters were identified and plotted for inspection. We then plotted the expression levels of a set of candidate genes (features) to identify the cluster representing the PMd and to differentiate it from the lateral and medial premammillary nuclei (LM and MM). To confirm our cluster identification, we extracted markers for the identified PMd cluster as well as differential markers for the PMd cluster vs. MM and LM clusters. Qualitative comparison of these markers with in situ hybridization data from the Allen Mouse Brain Atlas confirmed PMd cluster identity. Within the PMd cluster, we then extracted and plotted the level of co-expression of *Cck* and *Foxb1* within each cell of the cluster.

## Data analysis and statistics

Statistical analysis and data handling was performed with custom written codes in R/RStudio and with Python in Jupyter Notebooks.

WBP data was first plotted as line plots and violin plots with the ggplot2 package within R. To check whether data matched the test assumptions, data was first assessed with a Shapiro-Wilks test and with visual inspection of Q-Q plots.

Before the analysis of the WBP data, outliers above and below 1.5 x the interquartile range were removed and the data was averaged across each condition (saline, clozapine, CNO). WBP experiments were analyzed by a 3x3 mixed-design ANOVA with Huynh-Feldt Sphericity correction, followed by post-hoc two-tailed paired Welch's t-Tests. Hedge's g with a correction for paired data (*Gibbons et al., 1993*) was calculated for all statistically significant pairwise comparisons.

The hot plate and open field experiments were analyzed with two-tailed paired Welch's t-Tests if assumptions of normality were met. Otherwise, Wilcoxon signed-rank tests were used. Statistical analysis was limited to the groups that contained at least three subjects. Hedge's g with a correction for paired data (*Gibbons et al., 1993*) was calculated for all statistically significant pairwise comparisons. Pose estimation data from open field experiments were further analyzed and plotted in R. A set of functions for the analysis of DeepLabCut labeled data (*Sturman et al., 2020*) was used and modified where necessary according to the analysis requirements. The open field arena was partitioned into arena, periphery, center, and corner zones according to the default parameters. Animal movement and zone visits were calculated with a movement cut-off of 1 and an integration period of 13 (with a video frame rate of 25 frames/second). For optogenetic experiments, a 3-min bin immediately before (baseline) and 3 min immediately after (stim) the start of the optogenetic stimulation were used for statistical comparison and plotting. For chemogenetic experiments, 5 min bins (following a 5-min habituation) were used from each experiment (2 x BL, 2 x Clo for each animal) for statistical comparison and plotting. The telemetrically recorded cardiovascular and movement-related data were analyzed in Ponemah. To be able to identify the LEDon and LEDoff periods in these data, two video cameras were used in parallel: camera 1 was recording the animal (photostimulation light visible), and camera 2 was recording a computer screen displaying the telemetric recording in real-time. To quantify the effect of the optogenetic activation (respectively of the change from LEDoff of LEDon in control animals), data of the last 60 s of the LEDoff condition were used as baseline data and statistically compared to the data of the first 60 s of the LEDon condition. Multivariate analysis of variance (MANOVA) was used to test for the presence of overall differences between the LEDoff vs. LEDon conditions. If significant, pair-wise ordinary least squares (OLS) was used for post-hoc analysis. To correct for multiple testing (four dependent variables), p=0.5/4=0.125 was set as threshold level for post-hoc significance (Bonferroni correction).

To assess heart rate variability (HRV), the heart rate data were analyzed in full, that is 1-epoch resolution. The duration of each individual heart cycle was derived by calculating the difference between the timestamp of each pair of two subsequent heart cycles. The mean, median and SD of all heart cycle durations during the last baseline minute (LEDoff), respectively during the first minute of the LEDon condition, were calculated. The SD of all heart cycle durations during that minute was used as an indicator for HRV.

## Acknowledgements

We thank Christiane Marti and Laurence Clément for their technical support. The first part of the plethysmography experiments was conducted by SN in the laboratory of Prof. Julian Paton, Physiology Dept., University of Bristol (UK), now at the University of Auckland, New Zealand. We also thank Prof. Dr. Matthias Hänggi, Intensive Care Unit, University Hospital Bern for lending us the telemetric receiver equipment. We are further grateful for the financial support provided by the Swiss National Science Foundation (SNSF grant 31003 A_160325).

## Additional information

### Funding

| Funder | Grant reference number | Author |
| --- | --- | --- |
| Swiss National Science Foundation | grant 31003A_160325 | Marco R Celio |

The funders had no role in study design, data collection and interpretation, or the decision to submit the work for publication.

### Author contributions

Reto B Cola, Conceptualization, Formal analysis, Investigation, Methodology, Writing – original draft, Writing – review and editing; Diana M Roccaro-Waldmeyer, Formal analysis, Investigation, Methodology, Writing – original draft; Samara Naim, Alexandre Babalian, Investigation, Methodology; Petra Seebeck, Methodology; Gonzalo Alvarez-Bolado, Resources, Writing – original draft; Marco R Celio, Conceptualization, Resources, Supervision, Funding acquisition, Investigation, Writing – original draft, Writing – review and editing

### Author ORCIDs

Reto B Cola ⓘ https://orcid.org/0000-0003-1419-2480
Gonzalo Alvarez-Bolado ⓘ http://orcid.org/0000-0002-3044-1603
Marco R Celio ⓘ http://orcid.org/0000-0002-3623-842X

### Ethics

All experiments were approved by the Swiss federal and cantonal committee for animal experimentation (2016_20E_FR and 2021-10-FR) and were conducted in accordance with the institutional guidelines of the University of Fribourg.

Reviewer #1 (Public Review): https://doi.org/10.7554/eLife.86737.3.sa1
Reviewer #2 (Public Review): https://doi.org/10.7554/eLife.86737.3.sa2
Author Response https://doi.org/10.7554/eLife.86737.3.sa3

## Additional files

### Supplementary files

• Supplementary file 1. Track visualizations with underlying density maps of all tested DREADD animals.
• Supplementary file 2. Zone visit diagrams of all tested DREADD animals.

- Supplementary file 3. Track visualizations with underlying density maps of all tested optogenetic animals.
- Supplementary file 4. Zone visit diagrams of all tested optogenetic animals.
- Supplementary file 5. Heart rate variability (HRV) during the optogenetic activation of the Foxb1-terminal endings in the dlPAG of mouse 106-21/10. Comparison between the one-minute period before (green rectangle) and after (red rectangle) the start of the optogenetic activation. While all 4 parameters changed significantly during optogenetic stimulation (heart rate, systolic BP, diastolic BP, mean BP), the SD of heart cycle duration changed by a factor of more than five, from 3.8ms during the baseline period to 20.6ms during optogenetic activation.
- Supplementary file 6. Antibodies and fluorescent labels used for histological processing of brain sections.
- MDAR checklist

## Data availability

Raw data generated in this study was deposited on Dryad at: https://doi.org//10.5061/dryad.rv15dv4fn.Custom written code generated for this publication are available on GitHub in the following repository: https://github.com/rbcola/Parvafox-eLife-2024, (copy archived at *Cola, 2024*).

The following dataset was generated:

| Author(s) | Year | Dataset title | Dataset URL | Database and Identifier |
|---|---|---|---|---|
| Cola RB, Roccaro-Waldmeyer DM, Naim S, Babalian A, Seebeck P, Alvarez-Bolado G, Celio MR | 2024 | Chemo- and optogenetic activation of hypothalamic Foxb1-expressing neurons and their terminal endings in the rostral-dorsolateral PAG leads to tachypnea, bradycardia, and immobility | https://doi.org//10.5061/dryad.rv15dv4fn | Dryad Digital Repository, 10.5061/dryad.rv15dv4fn |

The following previously published dataset was used:

| Author(s) | Year | Dataset title | Dataset URL | Database and Identifier |
|---|---|---|---|---|
| Mickelsen LE, Flynn WF, Springer K, Wilson L, Beltrami EJ, Bolisetty M, Robson P, Jackson AC | 2020 | Single cell RNA sequencing to classify molecularly distinct neuronal and non-neuronal cell types in the mouse ventral posterior hypothalamus | https://www.ncbi.nlm.nih.gov/geo/query/acc.cgi?acc=GSE146692 | NCBI Gene Expression Omnibus, GSE146692 |

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
