## [Editor Report · eLife assessment]

This paper describes **valuable** results from studies investigating circuits in the brain that underlie behavioral responses in fearful situations. The authors identified a role for a class of neurons that are sufficient to cause these stereotyped behaviors including freezing behaviors. These **solid** studies increase our understanding of brain pathways regulating these types of behaviors.

---

## [Referee Report · Reviewer #1 (Public Review)]

In this study, the authors examined the putative functions of hypothalamic groups identifiable through Foxb1 expression, namely the parvofox Foxb1 of the LHA and the PMd Foxb1, emphasizing innate defensive responses. First, they reported that chemogenetic activation of Foxb1hypothalamic cell groups led to tachypnea. The authors tend to attribute this effect to the activation of hM3Dq expressed in the parvofox Foxb1 but did not rule out the participation of the PMd Foxb1 cell group, which may as well have expressed hM3Dq, particularly considering the large volume (200 nl) of the viral construct injected. Notably, the activation of the Foxb1hypothalamic cell groups in this experiment did not alter the gross locomotor activity, such as time spent immobile state. Thus, this contrasts with the authors' finding on the optogenetic activation of the Foxb1hypothalamic fibers projecting to the dorsolateral PAG. In the second experiment, the authors applied optogenetic ChR2-mediated excitation of the Foxb1+ cell bodies' axonal endings in the dlPAG, leading to freezing and, in a few cases, bradycardia. The effective site to evoke freezing was the rostral PAGdl, and fibers positioned either ventral or caudal to this target had no response. Considering the pattern of Foxb1hypothalamic cell groups projection to the PAG, the fibers projecting to the rostral PAGdl are likely to arise from the PMd Foxb1 cell group and not from the parvofox Foxb1 of the LHA. Here, it is important to consider that activation of PMd CCK cell group, which consists of around 90% of the PMd cells, evokes escape, not freezing. According to the present findings, a specific population of PMd Foxb1 cells may be involved in producing freezing. In addition, only a few of the animals with correct fiber placement presented sudden onset of bradycardia in response to the photostimulation. Considering the authors' findings, the Foxb1+ hypothalamic groups are likely to mediate behavioral responses related to innate defensive responses, where the parvofox Foxb1 of the LHA would be involved in promoting tachypnea and the PMd Foxb1group in mediating freezing and bradycardia. These findings are exciting, and, at this point, they need to be tested in a scenario of actual exposure to a natural predator.

---

## [Referee Report · Reviewer #2 (Public Review)]

The authors aimed to examine the role of a group of neurons expressing Foxb1 in behaviors through projections to the dlPAG. Standard chemogenetic activation or inhibition and optogentic terminal activation or inhibition at local PAG were used and results suggested that, while activation led to reduced locomotion and breathing, inhibition led to a small degree of increased locomotion.

The observed effects on breathing are evident and dramatic. However, due to the circumstance that does not permit to perform additional experiments, the conclusion is not as strong as it could be.

---

## [Author Response]

The following is the authors’ response to the original reviews.

**Public Reviews:**

**Reviewer #1 (Public Review):**
In this study, the authors examined the putative functions of hypothalamic groups identifiable through Foxb1 expression, namely the parvofox Foxb1 of the LHA and the PMd Foxb1, with emphasis on innate defensive responses. First, they reported that chemogenetic activation of Foxb1hypothalamic cell groups led to tachypnea. The authors tend to attribute this effect to the activation of hM3Dq expressed in the parvofox Foxb1 but did not rule out the participation of the PMd Foxb1 cell group which may as well have expressed hM3Dq, particularly considering the large volume (200 nl) of the viral construct injected. It is also noteworthy that the activation of the Foxb1hypothalamic cell groups in this experiment did not alter the gross locomotor activity, such as time spent immobile state. Thus, contrasts with the authors finding on the optogenetic activation of the Foxb1hypothalamic fibers projecting to the dorsolateral PAG. In the second experiment, the authors applied optogenetic ChR2-mediated excitation of the Foxb1+ cell bodies' axonal endings in the dlPAG leading to freezing and, in a few cases, bradycardia as well. The effective site to evoke freezing was the rostral PAGdl, and fibers positioned either ventral or caudal to this target had no response. Considering the pattern of Foxb1hypothalamic cell groups projection to the PAG, the fibers projecting to the rostral PAGdl are likely to arise from the PMd Foxb1 cell group, and not from the parvofox Foxb1 of the LHA. Here it is important to consider that optogenetic ChR2-mediated excitation of the axonal endings is likely to have activated the cell bodies originating these fibers, and one cannot ascertain whether the behavioral effects are related to the activation of the terminals in the PAGdl or the cell bodies originating the projection.

Authors’ reply: We acknowledge and agree about the possibility of backpropagation in ChR2mediated terminal stimulation experiments. We have introduced a paragraph in the discussion section discussing this issue. In short, the observation of an opposing phenotype in ArchT3.0 animals indicates, that the ChR2-mediated phenotype is indeed Foxb1-PAG projection specific. This is due to the fact, that the use of light-activated proton pumps for terminal stimulation can not induce backpropagation of an inhibitory effect to the soma. Potential downsides of the use of proton pumps in small compartments as e.g. in the axon are also discussed.

Moreover, activation of PMd CCK cell group, which consists of around 90% of the PMd cells, evokes escape, and not freezing. According to the present findings, a specific population of PMd Foxb1 cells may be involved in producing freezing. In addition, only a small number of the animals with correct fiber placement presented sudden onset of bradycardia in response to the photostimulation. Considering the authors' findings, the Foxb1+ hypothalamic groups are likely to mediate behavioral responses related to innate defensive responses, where the parvofox Foxb1 of the LHA would be involved in promoting tachypnea and the PMd Foxb1group in mediating freezing and bradycardia. These findings are very interesting, and, at this point, they need to be tested in a scenario of real exposure to a natural predator.

Authors’ reply: We fully agree with the proposed experiments. Due to the previously mentioned retirement of Prof. Celio and the concomitant expiration of licenses for animal experimentation we are prevented from conducting these experiments on our own. We have integrated a statement in the discussion, regarding these potential future experiments.

**Reviewer #2 (Public Review):**
The authors aimed to examine the role of a group of neurons expressing Foxb1 in behaviors through projections to the dlPAG. Standard chemogenetic activation or inhibition and optogentic terminal activation or inhibition at local PAG were used and results suggested that, while activation led to reduced locomotion and breathing, inhibition led to a small degree of increased locomotion.The observed effects on breathing are evident and dramatic. However, this study needs significant improvements in terms of data analysis and presentation and some of studies seem incomplete; and therefore the data may not yet support the conclusion.1. Fig.1 has no experimental data and needs to be replaced with detailed pictures from the viral injected mice showing the projections diagrammed.

Authors’ reply: We believe that this graphic illustration is helpful to the reader to comprehend the spatial relationship between the parvafoxFoxb1 nucleus, the mammillary nuclei, and the PAG. In a previous study we have characterized the projections of the parvafoxFoxb1 nucleus in detail (using the same Foxb1-Cre mouse line as in the present study) and, in this regard, would like to refer Reviewer #2 to this publication (https://onlinelibrary.wiley.com/doi/10.1002/cne.24057).

1. Fig. 3 needs control pictures and statistical comparison with different conditions in c-Fos. Also expression in other nearby regions needs to be presented to demonstrate the specificity of the expression.

Authors’ reply: We have modified the original Fig. 3 with more pictures across all three conditions used in the chemogenetic experiments. Since the new figure now takes up a whole page, and because the data in this figure is for validation purpose of the DREADD experiments, we have decided to rather put it into the supplementary files. The figure is now labelled as “Supplementary File S1”. All figure and file numberings throughout the text have been adjusted accordingly.

1. Fig. 5, a great effort has been made to illustrate the point that CCK and Foxb1 are differentially expressed. Why not just perform a double in situ experiment to directly illustrate the point?

Authors’ reply: We have addressed this comment in the initial release of the eLife manuscript. In short, we agree that a double ISH experiment would have been an alternative approach, but would like to state that scRNAseq is a well established and valid method for this purpose.

1. Fig. 7 data on optogenetic stimulation on immobility and breathing, since not all mice showed the same phenotype, what is the criterion for allocating these mice to hit or no hit groups? Given the dramatically reduced breathing and locomotion, what is the temperature response? More data needs to be gathered to support that this is a defense behavior.

Authors’ reply: The criteria for allocation of animals to the experimental groups is described in section “Optogenetic modulation of Foxb1 terminals in the dlPAG induces immobility” and is based on the stereotaxic coordinates of the tips of the glass fiber implants. We did not perform any experiments, in which we recorded body temperatures or temperature preferences in optogenetic animals. Such experiments were outside the scope of the study. As mentioned in a previous comment above, we have added an additional paragraph to the discussion section regarding future investigations of these hypothalamic Foxb1 neurons during exposure to natural predators. Such experiments would certainly allow more insight into the defensive nature of the described phenotype.

1. The authors claim to target dlPAG. However, in the picture shown in Fig. 8C, almost all PAG contains ChR2 fibers and it is likely all the fibers will be activated by light. Thus, as presented, the data does not support the claim of the specificity on dlPAG. Also c-Fos data needs to be presented on the degree of activation of downstream PAG neurons after light exposure.

Authors’ reply: We attach the original image 8c, without arrows and indications, in which the localization of ChR2-positive fibers in the dlPAG is better visible. They are located exactly under the tip of the fiberoptic fiber. We do not know the functional characteristics of the post-synaptic PAG neurons and have not determined experimentally their downstream targets. Investigating the downstream target was outside the scope of the current publication.

1. Fig. 9 only showed one case. A statistical comparison needs to be presented.

Authors’ reply: Our cardiovascular experiments are of exploratory and descriptive nature (i.e. pilot experiments). It was a conscious decision to not perform hypothesis tests on these experiments. We did not have enough mice to perform statistical tests with sufficient statistical power. Providing results from hypothesis tests on these data would lead to statistically unjustified conclusions. To clarify this issue, we have added a paragraph to the relevant results section.

1. Optogentic terminal activation in the PAG will likely elicit back-propagation and subsequent activation of additional downstream brain sites of Foxb1 neurons. More experiments need to be done to assess this and as presented, the data does not support the role of PAG necessarily.

Authors’ reply: Please see our answer to Reviewer #1 regarding the same issue.

1. The authors claim negative data from PVH-Cre mice. More data need to be presented to make this case.

Authors’ reply: We would like to refer to our answer to point 6 that was raised by Reviewer #2

The conclusion, even as presented, adds to the known evidence of the PAG in the defense behavior.
**Reviewer #1 (Recommendations For The Authors):**
In the pharmacogenetic experiments, the authors need to clarify which Foxb1hypothalamic presented the activation of hM3Dq. It is important to know whether this activation-producing tachypnea was restricted to the parvofoxFoxb1 or also included the PMd Foxb1 group. It would be important to isolate the effect of the pharmacogenetic activation of each one of these Foxb1 hypothalamic cell groups.After determining which cell group would be involved in mediating this respiratory effect, it would be nice to discuss the possible pathways involved in this effect.In the optogenetic experiments, the authors should differentiate between the effects of the PAG projecting fibers from the PMd and those from the parvofox groups. As it stands, it seems that the freezing and bradycardia depend on projection from the PMd Foxb1 group to the rostral PAGdl. However, considering the large volume (200 nl) of the viral construct injected, both groups were likely to express channelrhodopsin, and it would be important if the authors could restrict the viral injections to each one of the Foxb1 hypothalamic cell groups.

Authors’ reply: We fully agree with the suggestion, but due to the recent retirement of Prof. Celio we unfortunately not allowed to conduct any further animal experiments.

The authors also reported that photoactivation ventral to the PAGdl, possibly in the PAGl did not yield any clear behavioral response. However, as pointed out in the discussion, a recent publication found that the parvofox Foxb1 projection to the lateral PAG drives social avoidance, and we were wondering whether there was any avoidance behavior during the photoactivation of the PAGl fibers.

Authors’ reply: We did not conduct any social avoidance experiments ourselves. However, we did perform ultrasonic vocalization experiments (unpublished data) in which we optogenetically stimulated Foxb1+ terminals in the PAG. Due to experimental issues related to the age of the tested mice, we did not obtain conclusive results regarding the ultrasonic vocalizations. By a purely observational account, we did not observe any active avoidance during optogenetic stimulation, but rather a cessation of interaction. We are unable to judge whether this was more pronounced in the PAGl targeted mice or not.

Another important point is that optogenetic ChR2-mediated excitation of the axonal endings is likely to activate the cell bodies originating these fibers, and one cannot ascertain whether the behavioral effects depend on the activation of the terminals in the PAGdl or the activation of the cell bodies originating these terminals. Note, in the present case, PMd cell bodies may also project elsewhere, such as the cuneiform nucleus, known to mediate freezing responses. To circumvent this problem, during photoactivation of the PAGdl terminals, the authors should inhibit the cell bodies originating these terminals.

Authors’ reply: We would like to refer to the answer we provided above regarding the issue of backpropagation or ChR2-mediated phenotypes and projection-specificity.

Another important issue is related to the fact that around 90% of the PMd express CCK (Wang et al., 2021), and previous work showed that activation of these cells yielded escape and not freezing (Wang et al., 2021). Although the authors claim that the single-cell RNA sequencing dataset reveals distinct Foxb1 expression in the PMd, these results derive from tissues collected in the posterior hypothalamus, not exactly restricted to the PMd. Therefore, it would be desirable if the authors could show CCK and Foxb1doulbe labeled PMd sections to evaluate the exact percentage of cells expressing either one of these peptides.

Authors’ reply: The tissues for the scRNAseq data were obtained from hypothalamic tissues between stereotaxic coordinates of AP-2.54 to AP-3.16 (please see Fig. 1b in Mickelsen et al. 2020) and not purely from the posterior hypothalamic nucleus. These tissues hence include a large proportion of the PMd neurons. We would like to point out that the expression profile of the PMd cluster matches well with the ISH data from the Allen Brain Atlas that we have put together in "Supplementary File S6” (originally “Supplementary File S5”)

The authors should also explain why only a small number of animals that received PAGdl photoactivation presented bradycardia. Moreover, they should also discuss the possible pathways mediating this effect. Here, it is important to point out that the cuneiform nucleus, as suggested by the authors as one possible way to mediate this effect, promotes sympathetic vasomotor activity (Verbene, 1995).

We have added the sentence: “The projections of the cuneiform nucleus to the rostral ventrolateral medulla promote sympathetic vasomotor activity (Verberne 1995).” to the Discussion section.

**Reviewer #2 (Recommendations For The Authors):**
In this reviewer's view, this study needs substantial improvement:1. The writing is very sloppy and difficult to follow. There is no clear logic flow in the main text and the figures need substantial realigning for panels, additions of labelling etc.

We have added the sentence.

1. Fig. 6 the hot plate data is out of place and should be placed in supplementary or removed completely.

Authors’ reply: We and others have previously shown that the parvalbumin+ population of the Parvafox nucleus is involved in nociceptive behavior. Hence, we believe it is of interest to show, that we do not see the same phenotype with the stimulation of the Foxb+ population of the parvafox nucleus. This data shows that the nociceptive component of the parvafox nucleus is confined to its parvalbumin+ population.

1. The authors discussed social behavior data in the Discussion, but no such data is presented, which is very confusing.

Authors’ reply: Indeed we did not perform any experiments to investigate social behavior. However, we address that the observed locomotive phenotype of optogenetic Foxb1+-terminals could have lead to a bias in the interpretation of the social behavior experiments published elsewhere by others.

1. The authors discussed a great deal on potential differences between parvafox and PMd Foxb1 neurons, however, no clear data was presented to show a functional difference between them, which is also confusing.

Authors’ reply: Even though investigations on the functional differences of parvafox and PMd Foxb1 neurons would be highly interesting, it was outside the scope of the current study. Due to the recent retirement of Prof. Celio, we are not allowed to perform any additional animal experiments.